# Progressive mural cell deficiencies across the lifespan in a *foxf2* model of cerebral small vessel disease

**Merry Faye E Graff[1,2], Emma EM Heeg[1,2], David A Elliott[3], Sarah J Childs[1,2]\***

[1]Alberta Children's Hospital Research Institute, University of Calgary, Calgary, Canada; [2]Department of Biochemistry and Molecular Biology, University of Calgary, Calgary, Canada; [3]Hotchkiss Brain Institute Advanced Microscopy Platform, University of Calgary, Calgary, Canada

## eLife Assessment

This study provides **important** insights into mural cell dynamics and vascular pathology using a zebrafish model of cerebral small vessel disease. The authors present **convincing** evidence that partial loss of foxf2 function results in progressive, cell-autonomous defects in pericytes accompanied by endothelial abnormalities across the lifespan. By leveraging advanced in vivo imaging and genetic approaches, the work establishes zebrafish as a powerful and relevant model for dissecting the cellular mechanisms underlying cerebral small vessel disease.

**\*For correspondence:**
schilds@ucalgary.ca

**Competing interest:** The authors declare that no competing interests exist.

**Abstract** Cerebral small vessel disease (SVD) is a leading cause of stroke and dementia and yet is often an incidental finding in aged patients due to the inaccessibility of brain vasculature to imaging. Animal models are important for modelling the development and progression of SVD across the lifespan. In humans, reduced *FOXF2* is associated with an increased stroke risk and SVD prevalence in humans. In the zebrafish, *foxf2* is expressed in pericytes and vascular smooth muscle cells and is involved in vascular stability. We use partial *foxf2* loss of function (*foxf2a⁻/⁻*) to model the lifespan effect of reduced Foxf2 on small vessel biology. We find that the initial pool of pericytes in developing *foxf2a* mutants is strongly reduced. The few brain pericytes present in mutants have strikingly longer processes and enlarged soma. *foxf2a* mutant pericytes can partially repopulate the brain after ablation, suggesting some recovery is possible. Despite this capacity, adult *foxf2a* mutant brains show regional heterogeneity, with some areas of normality and others with severe pericyte depletion. Taken together, *foxf2a* mutants fail to generate a sufficient initial population of pericytes. The pericytes that remain have abnormal cell morphology. Over the lifespan, initial pericyte deficits are not repaired and lead to severely abnormal cerebrovasculature in adults. This work opens new avenues for modeling progressive genetic forms of human cerebral small vessel disease.

## Introduction

Brain microvessels supply billions of neurons and other brain cells with the oxygen and nutrients they need. Pathologies affecting brain microvasculature progress slowly and silently over a lifetime but have devastating consequences from decreased perfusion and vascular destabilization. Cerebral small vessel disease (CSVD) encompasses progressive heterogeneous changes in brain microvessels; it is the most common cause of vascular dementia and a significant contributor to stroke and cognitive decline (Østergaard et al., 2016). 25% of all strokes are the result of CSVD, yet effective targeted

**eLife digest** Every time you pause to think, remember a name, or read a sentence, the blood in your brain is quickly rerouted to the neurons doing the work. This redistribution depends on a vast network of blood vessels, from large arteries to microscopic capillaries, which deliver oxygen and energy directly to active brain cells.

For this system to function properly, the smallest blood vessels, the capillaries, must be able to regulate blood flow precisely. This control is provided by support cells on the outside of the capillary, such as pericytes or smooth muscle cells, which relax to open the vessel. When these cells fail, brain regions may no longer receive enough blood, even if larger vessels remain intact.

A breakdown of these cells is observed in cerebral small vessel disease, a leading cause of stroke and dementia. Unlike other types of strokes, this disease originates in the smallest blood vessels of the brain. However, it remains unclear whether it begins only in old age or much earlier in life. Understanding when and how this disease progresses is important because identifying its earliest mechanisms may offer opportunities to delay damage.

Graff et al. studied a zebrafish model carrying a mutation in foxf2a, which is linked to cerebral small-vessel disease in older humans. They found that the condition may not be exclusively age-related. When zebrafish had foxf2a levels reduced to about 50% of normal - similar to the reduction observed in humans with variants linked to cerebral small vessel disease - the fish developed blood vessel absormalities from the earlierst stages of life that persisted into adulthood. They also had fewer pericytes. Although pericytes could regenerate to some extent, blood vessel damage remained and worsened over the lifespan in this zebrafish model.

More detailed analyses revealed that pericytes showed signs of stress, which caused higher rates of cell death compared to zebrafish with normal foxf2a levels. In other words, although blood vessel damage could be partly repaired, it tended to deteriorate when foxf2a was absent.

These findings suggest that cerebral small vessel disease should may be better understood as a lifelong, progressive condition, where damage accumulates over time. Although approximately 20% of the population may carry genetic risk factors for this kind of disease, ongoing blood vessel damage and repair are common. Population-wide screening for individuals at risk of cerebral small vessel disease early in life, combined with targeted lifestyle and cardiovascular interventions, could greatly reduce the disease burden in the elderly.

---

treatments remain elusive (*Østergaard et al., 2016*). This is in part due to the inability to both detect and assess progressive damage in the brain.

While there are several genes implicated in familial CSVD (i.e. *NOTCH3*, *HTRA1*, *FOXC1*, *COL4A1*, and *COL4A2*), there is a lack of suitable in vivo models for studying disease development and progression. Research has predominantly focused on *NOTCH3*, but over the last decade, *FOXF2* has emerged as a risk locus for CSVD (*Chauhan et al., 2016*; *Duperron et al., 2023*). SNPs in the intergenic region between *FOXF2* and *FOXQ1* decrease *FOXF2* expression and significantly increase stroke risk due to the variant decreasing the efficiency of *ETS1* binding to a novel *FOXF2* enhancer (*Ryu et al., 2022*).

Foxf2 promotes mural cell differentiation and vascular stability in the zebrafish brain and is expressed highly in brain pericytes (*Ahuja et al., 2024* #1591); (*Chauhan et al., 2016* #1403); (*Reyahi et al., 2015* #1372); (*Ryu et al., 2022* #1590). Brain pericytes interact closely with endothelial cells, contributing to extracellular matrix (ECM) deposition and blood-brain barrier (BBB) formation, in addition to providing vasoactivity and stability (*Bahrami and Childs, 2020*; *Daneman et al., 2010*; *Dave et al., 2018*; *Stratman et al., 2009*). In animal models, an absence of brain pericytes results in hemorrhages and accelerates vascular-mediated neurodegeneration (*Bell et al., 2010*; *Wang et al., 2014*). Foxf2 is clearly important for vascular stability across species, as loss of Foxf2 in mice and zebrafish leads to increased brain hemorrhage and alterations in brain pericyte numbers and differentiation (*Chauhan et al., 2016*; *Reyahi et al., 2015*; *Ryu et al., 2022*). Brain tissue from patients with aging-related dementias (i.e. post-stroke dementia, vascular dementia, Alzheimer's disease) has reduced deep white matter pericytes and associated BBB disruption (*Ding et al., 2020*), suggesting that pericytes should be examined as mediators of CSVD progression in patients with FOXF2 deficiency.

We previously showed that complete loss of *foxf2* in *foxf2a*^ca71^; *foxf2b*^ca21^ double-homozygous mutants in late embryogenesis leads to reduced brain pericyte numbers (*Ryu et al., 2022*). However, stroke susceptibility in humans is associated with reduced, but not absent, FOXF2 expression. Genome-wide association (GWA) indicates that carrying a minor allele of an SNP in a *FOXF2* enhancer leads to reduced, but not absent, FOXF2 and is associated with stroke (*Ryu et al., 2022*). For this reason, we model CSVD using a zebrafish with reduced Foxf2 dosage using single homozygous *foxf2a* mutants. Zebrafish *foxf2a* and *foxf2b* genes are the result of genome duplication in zebrafish ~430 million years ago and have similar gene expression (*Arnold et al., 2015*; *Chauhan et al., 2016*). We have detected no difference in function between the two genes and, therefore, *foxf2a* loss of function may be similar to human heterozygous loss of FOXF2 function, a state that is observed in the population in GnomAD (*Chen et al., 2024*).

Strikingly, while pericytes in embryonic *foxf2* mutants are clearly affected, *foxf2* mutants can survive until adulthood, albeit with a reduced lifespan. How pericytes change across the lifespan while CSVD progresses is unknown. Here, we find that *foxf2a* mutants have significantly reduced brain pericyte numbers as embryos that do not recover over time. Pericytes in mutant embryos and larvae exhibit morphological abnormalities, including increased soma size, longer processes, and degeneration. We show that processes and soma in the adults are also abnormal, though their morphology differs over the lifespan. Although the initial pool of pericytes is smaller, mutants can regenerate pericytes after ablation. Our analysis suggests that *foxf2* is required within pericytes to modulate numbers but also has a strong effect on morphology. We show that brain pericytes may contribute to the pathological progression of genetic CSVD, starting in embryonic development and continuing across the lifespan. Understanding the early developmental aspects of late-onset vascular conditions like CSVD will aid in the development of effective therapeutic strategies.

## Results

### Pericyte number is consistently lower in foxf2 mutant embryos and larvae

Embryonic phenotypes can lead to lifelong consequences. We have previously studied *foxf2a;foxf2b* double mutants only at a single embryonic stage at 3 days post-fertilization (dpf). To understand how a pericyte and cerebrovascular phenotype evolves and/or resolves over development, we conducted serial imaging of individual brains of *foxf2a* mutants at embryonic stages (3 and 5 dpf), and at larval stages (7 and 10 dpf). Mutants were live imaged using endothelial (*kdrl:mCherry*) and pericyte (*pdgfrβ:Gal4, UAS:GFP*) transgenic lines.

Embryonically, brain pericytes have a thin-strand morphology and are closely associated with, and extend processes over, the endothelium in the midbrain and hindbrain of zebrafish (*Figure 1A–A″*). In wild-type embryos, the number of pericytes increases progressively from 3 through 10 dpf (*Figure 1B*). However, *foxf2a* mutants show significantly fewer pericytes on brain vessels at 3 dpf (*Figure 1C*; mean 21 in wild-type and 10 in mutants), and this pericyte deficiency persists through 5, 7, and 10 dpf (*Figure 1C–D*). The reduction in pericyte numbers shows variable penetrance, with some *foxf2a* mutants having pericyte numbers only slightly reduced from wild-type, and others that are severely diminished. The same pattern of pericyte reduction is seen with *foxf2a* mutants from a homozygous or heterozygous incross, suggesting there is no maternal effect (*Figure 1E–F*). Serial imaging of mutants with regional absence in earlier stages shows that defects in pericyte coverage persist into later stages, suggesting that the size of the initial pericyte population is a key determinant of later coverage (*Figure 1—figure supplement 1*). Double *foxf2a;foxf2b* mutants have a fully penetrant phenotype with significantly fewer brain pericytes in mutants than wild-type at every stage (*Figure 1—figure supplement 2*; mean 19 in the wild-type and 9 in mutants at 3 dpf). Incomplete penetrance in *foxf2a* single mutants could be due to genetic compensation from *foxf2b*. While *foxf2b* is not significantly upregulated in *foxf2a* mutants on average, individual embryos have highly variable *foxf2b* expression (*Figure 1—figure supplement 3*).

Since *Foxf2* conditional knockout mice show reduced expression of the pericyte marker *Pdgfrβ*, we tested whether *pdgfrβ* expression is *reduced* in *foxf2a* mutants, as this might introduce inaccuracies in pericyte counting. We used two methods, quantitative hybridization chain reaction (HCR) in situ hybridization and integrated density of the *pdgfrβ* transgene expression in wild-types and mutants

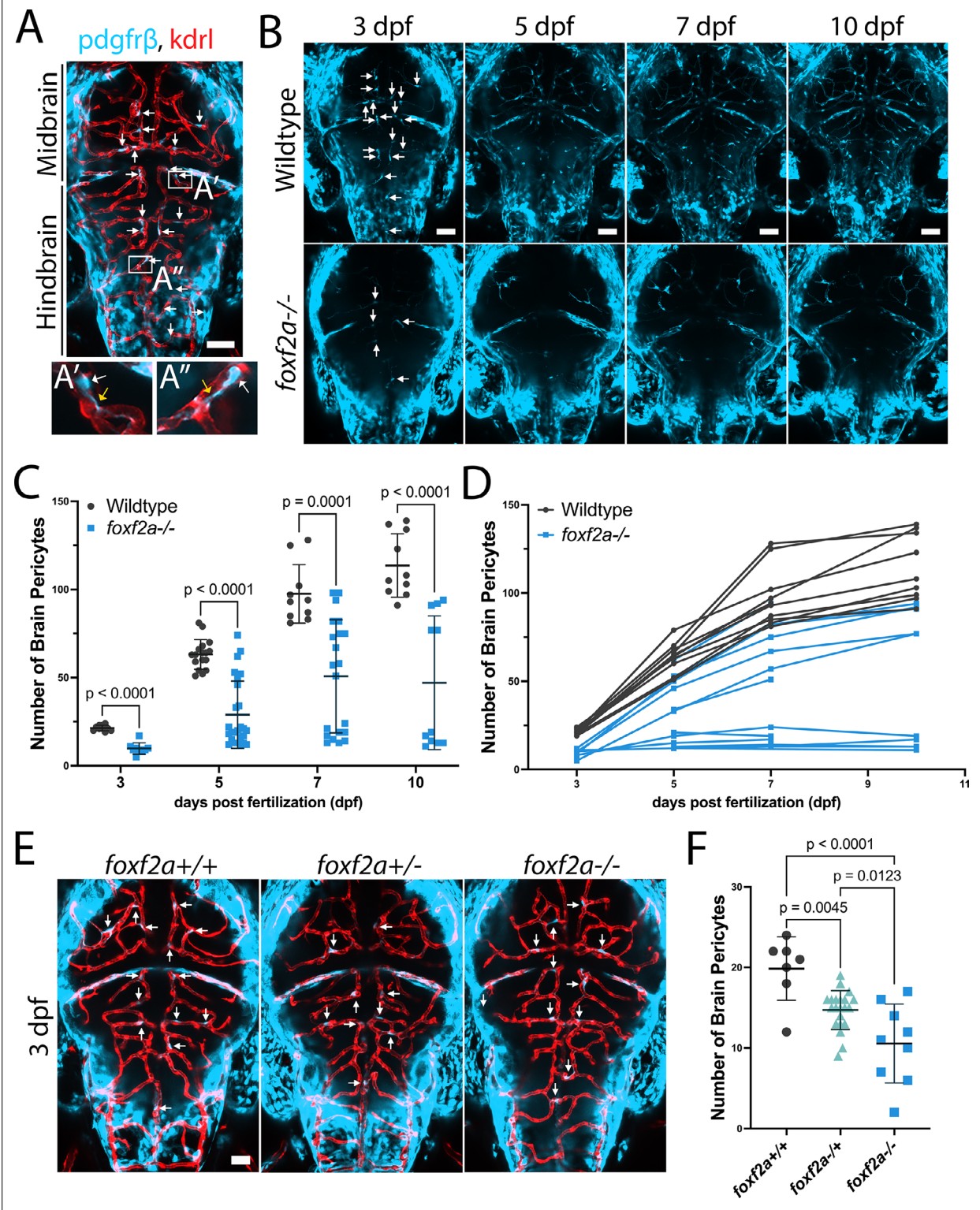

**Figure 1.** Brain pericyte number is consistently lower and does not recover in *foxf2a* mutant larvae. (**A**) Zebrafish brains were imaged using endothelial (red; *Tg(kdrl:mCherry)*) and pericyte (light blue; *Tg(pdgfrβ:Gal4, UAS:GFP)*) transgenic lines (arrows: brain pericytes). (**A'-A"**) Brain pericyte soma (white arrows) and processes (yellow arrows) are closely associated with the endothelium. (**B**) Serially imaged wild-type and *foxf2a* mutant brains at 3, 5, 7, and 10 dpf. (**C**) Total brain pericyte numbers at 3, 5, 7, and 10 dpf. (**D**) Individual brain pericyte trajectories of serially imaged embryos over the same period. (**E**) Dorsal images of embryos for the indicated genotypes from a *foxf2a* heterozygous incross at 75 hpf. (**F**) Total brain pericytes at 75 hpf. Statistical analysis was conducted using multiple Mann-Whitney tests (**C**) and one-way ANOVA with Tukey's test (**F**). Scale bars, 50 µm (**A–B, E**).

*Figure 1 continued on next page*

*Figure 1 continued*

The online version of this article includes the following source data and figure supplement(s) for figure 1:

**Source data 1.** All raw quantitative data underlying *Figure 1* and supplements.

**Figure supplement 1.** *foxf2a* mutants exhibit regional loss.

**Figure supplement 2.** foxf2 knockouts have severe pericyte deficiency during development.

**Figure supplement 3.** *pdgfrβ* mRNA and transgene have similar expression in wild-type and *foxf2a* mutants.

carrying only a single copy of the transgene. We show that *pdgfrβ* mRNA nor transgene expression is not reduced in *foxf2a* mutants (*Figure 1—figure supplement 3*). Furthermore, we show that *pdgfrβ* has complete overlap in expression with other brain pericyte markers, such as *ndufa4l2* and *foxf2b* using HCR (*Figure 2A–D*). Thus, *pdgfrβ* transgene or mRNA expression can reliably be used to count zebrafish brain pericytes in *foxf2a* mutants.

Although we focus on mural cells, Foxf2 is expressed in adult mouse brain endothelium (*Vanlandewijck et al., 2018*). We assessed *foxf2a* and *foxf2b* mRNA expression patterns using HCR in situ hybridization. At 3 dpf, *foxf2a* is co-localized with the pericyte marker *ndufa4l2a* in brain pericytes and shows low-level co-localized expression with the blood vessel marker *kdrl* in brain endothelium (*Figure 2A*). The expression pattern of *foxf2b* is similar. It is expressed in pericytes (*pdgfrβ*) and only weakly in endothelial cells (*kdrl*) (*Figure 2B*). This is supported by the DanioCell atlas of single-cell sequencing of multiple embryonic stages that shows expression of both genes is strongest in mural cells, pericytes, and vascular smooth muscle cells and low in endothelial cells (*Sur et al., 2023*; *Figure 2—figure supplement 1*). Thus, while *foxf2a* and *foxf2b* are principally expressed in pericytes, they are lowly expressed in endothelium during brain development.

Pericyte loss or impairment leads to alterations in vascular patterning in the mouse retina (*Eilken et al., 2017*). To assess if the endothelial network is affected in *foxf2a* mutants, we employed a Python workflow using Vessel Metrics (*McGarry et al., 2024*; *Figure 2E*). We found no statistical difference in total network length between wild-type and mutants at 3 dpf (*Figure 2F*) or hindbrain central artery diameter (*Figure 2G*). However, pericyte density (number of pericytes divided by the total network length) is reduced by 40% in *foxf2a* mutants (*Figure 2H*), reflecting the loss of pericytes with no change in vessel network length. Similarly, pericyte coverage of vessels (total process coverage from brain pericytes divided by the total network length) is reduced by 39% in mutants (*Figure 2I*). Our data suggest that during early development, *foxf2a* depletion primarily affects pericytes.

## Early defects in brain vessel development have lifelong consequences

*foxf2* mutant animals can survive to adulthood, albeit with a reduced lifespan (~1 year vs. >2 years). Are early pericyte deficiencies repaired, or is loss of pericytes unimportant to survival to adulthood? To understand how brain pericyte phenotypes evolve over the lifespan, we dissected adult wild-type and *foxf2a* mutant brains on pericyte and endothelial double transgenic backgrounds and imaged after iDISCO clearing. Gross measurements of standard length of the fish (snout to tail) show no significant differences between wild-type and mutants, except that female mutant brain length and width are significantly smaller at 11 months post fertilization (mpf) (*Figure 2—figure supplement 2*). However, there is no significant difference in the proportional brain length/standard length ratio in *foxf2a* mutants vs. wild-type.

Projected 3D views of light sheet images of the whole brain show striking defects in pericyte density, coverage, and vascular pattern in *foxf2a* mutants vs. wild-type adults (*Videos 1 and 2*; *Figure 3A–B*). Pericyte distribution is irregular, and blood vessel density is visibly reduced in mutant brains (*Figure 3C–D*). Using a machine learning workflow (*Figure 3—figure supplement 1*, *Figure 3—source data 1*), we segmented pericyte soma, or the vessel backbone for the entire adult brain, to count the total pericyte number in comparison to the total endothelial network length. We find a significant reduction in pericyte numbers in 3 mpf *foxf2a* mutant brains, which have only 45% of the pericytes of a wild-type brain (average of 24,567 pericytes/brain vs. wild-type 54,833 pericytes/brain; n=3 of each genotype; *Figure 3E*). However, at this stage, the vessel network length is not statistically different (*Figure 3F*). The density of pericytes on vessels is significantly decreased from 0.01 pericyte/μm to 0.006 pericytes/μm in *foxf2a* mutants showing a clear deficit (*Figure 3G*). In contrast, the mean vessel diameter across vessel segments is not significantly different at 3 mpf when all diameters are

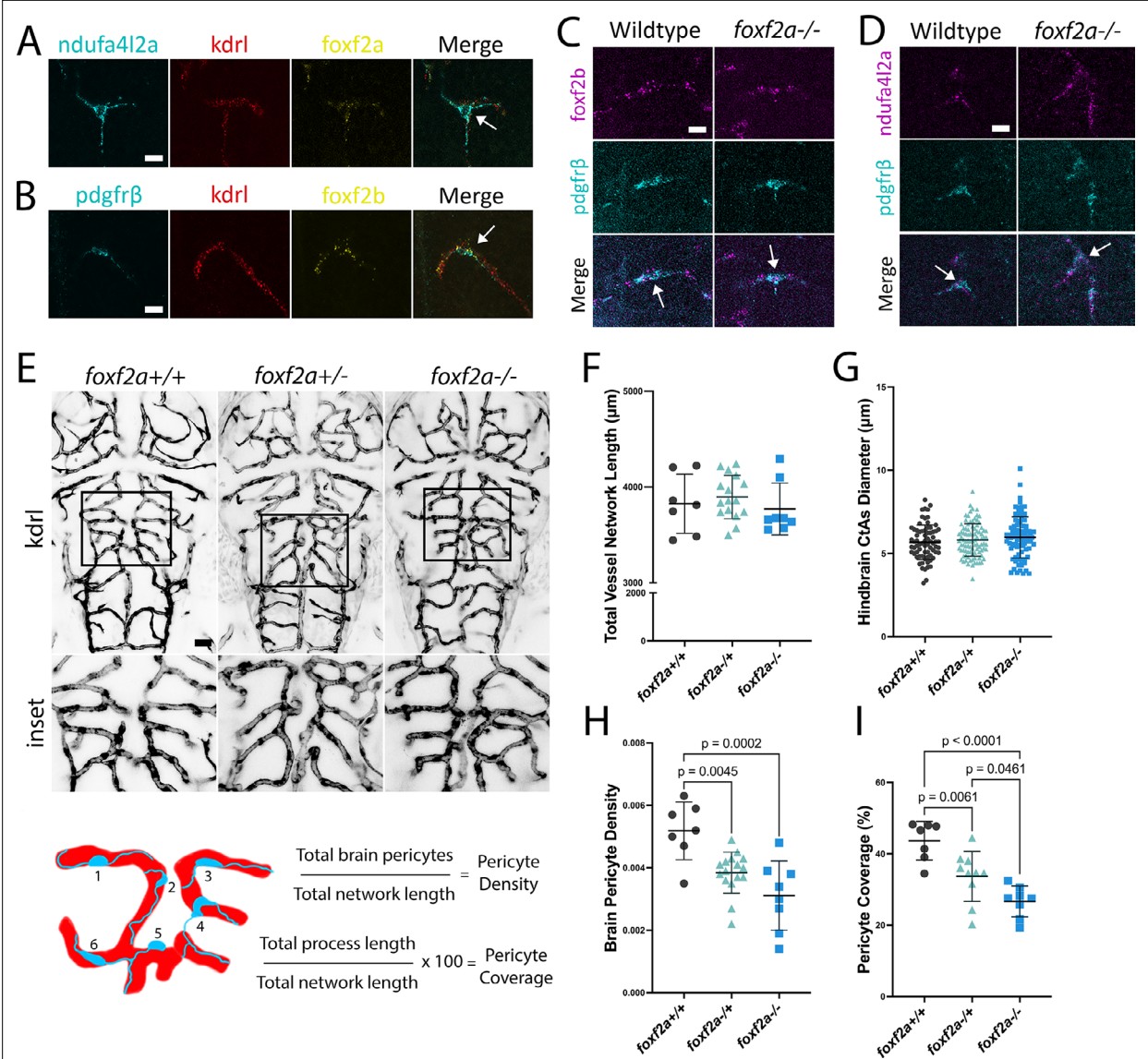

**Figure 2.** *Loss of foxf2* affects embryonic pericyte numbers, but not endothelial cell pattern. (**A**) *foxf2a* expression in wild-type brains at 72 hpf using hybridization chain reaction (HCR) shows co-expression with pericyte marker *ndufa4l2a*. *foxf2a* is also lowly co-expressed in the endothelium with *kdrl*. Arrows show overlapping expression. (**B**) *foxf2b* is co-expressed with pericyte marker *pdgfrβ,* also lowly expressed in the endothelium (*kdrl*). (**C**) *foxf2b* and *pdgfrβ* are expressed in a similar, overlapping pattern in pericytes of wild-type and *foxf2a* mutants. (**D**) Pericyte marker *nduf4al2a* and *pdgfrβ* are expressed in a similar, overlapping pattern in pericytes in wild-type and *foxf2a* mutants. (**E**) Image of endothelium used to generate the total blood vessel network length. (**F**) Total vessel network length from Vessel Metrics software. (**G**) Scatter plot of hindbrain CtA diameters. (**H**) Scatter plot of pericyte density and pericyte coverage (**I**). Statistical analysis was conducted using one-way ANOVA with Tukey's test. Scale bars, 10 μm (**A–D**), 50 μm (**F**).

The online version of this article includes the following source data and figure supplement(s) for figure 2:

**Source data 1.** All raw quantitative data underlying *Figure 2* and supplements.

**Figure supplement 1.** Expression of *foxf2a* and *foxf2b* in single-cell sequencing data from Daniocell.

**Figure supplement 2.** *foxf2a mutant* adult brains have normal size as compared to wild-types.

considered, nor when vessel diameters are grouped in 5 μm bins (n=522,037 wild-type and 381,544 *foxf2a* mutant diameters; *Figure 3H–I*).

Similarly, cleared dissected brains at 11 mpf were imaged and showed similarly striking vascular defects (*Figure 3—figure supplement 2*). Adult pericytes have a clear, oblong cell body with long, slender primary processes that extend from the cytoplasm and secondary processes that wrap around the circumference of the blood vessel (*Figure 3J*). These pericytes form a continuous network of

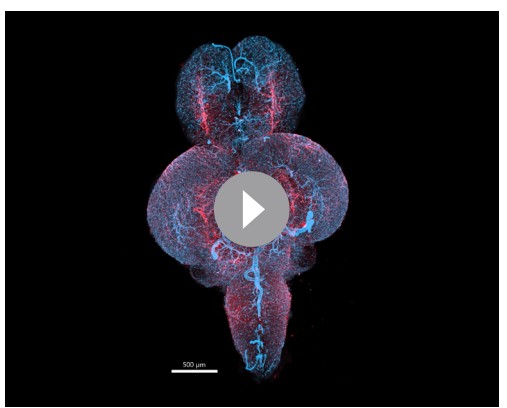

**Video 1.** Rotating view of a cleared wild-type brain at 3 mpf with *pdgfr*β (blue) and *kdrl* (red).
https://elifesciences.org/articles/106720/figures#video1

processes to cover the blood vessels in the brain. In *foxf2a* 11 mpf brains, pericyte somas lose their oblong shape, and cell bodies cannot be easily distinguished from processes. Mutant cells also extend thickened linear processes, with no secondary processes encircling the vessels. Mutant pericytes do not form an extensive network with other neighbouring pericytes. We note that there is variable phenotypic penetrance in *foxf2a* adults at 3 mpf that is reflective of the incomplete penetrance in *foxf2a* embryos (*Figure 3—figure supplement 2*).

To view cellular morphology, we sectioned adult brains and immunolabeled pericytes and endothelium. In wild-type adult brains, we identified three subtypes of pericytes: ensheathing, mesh, and thin-strand, previously characterized in murine models (*Berthiaume et al., 2018b*; *Figure 3—figure supplement 3*). In comparing brain sections from both wild-type and *foxf2a* mutants, on smaller vessels, mutant pericytes exhibit more linear processes with barely distinguishable soma, markedly differing from the characteristic appearance of wild-type adult pericytes (*Figure 3—figure supplement 4*). Similarly, *pdgfrβ*-expressing cells on large-calibre vessels (mural cells, likely vascular smooth muscle cells (vSMCs)) show alterations in morphology and coverage (*Figure 3—figure supplement 4*). Although mutant embryos do not exhibit apparent abnormalities in their endothelium, large aneurysm-like structures are evident in the adult brain (*Figure 3—figure supplement 5*). These structures also appear to have decreased *kdrl* expression. Thus, both whole-mount and sectioned tissue show that brain vascular mural cell number and morphology are severely impacted in adult *foxf2a*$^{-/-}$ mutant brains, with increasing involvement of the endothelium, suggesting a worsening phenotype throughout life.

As *foxf2a* is expressed in vSMCs, we tested the effect of loss of *foxf2a* on vSMCs using confocal imaging of larvae and light sheet microscopy of cleared dissected 6 mpf adult wild-type and *foxf2a* mutant brains stained for transgenic endothelial (*kdrl*) and vSMC (*acta2*) markers. *acta2*-positive vSMCs are present on brain vessels around the Circle of Willis at embryonic (5 dpf) and larval (10 dpf) stages in both mutant and wild-type (*Figure 4A and C*). Furthermore, the number of vSMCs does not differ between mutant and wild-type (*Figure 4B and D*; *Figure 4—source data 1*). In adult brains, we find no significant difference in the total length of vSMCs in the brain, or in vSMC coverage (proportion of vessels covered by vSMCs) (n=3 of each genotype; *Videos 3 and 4*; *Figure 4E–H*). We note that at 6 mpf, there is a significant decrease in vessel network length, however (*Figure 4G*).

## Morphological abnormalities emerge in the larval brain of mutants

Considering the altered pericyte morphology of adult mutants, we revisited developmental stages to identify when these abnormalities first arise. We analyzed morphology at 3 and 10 dpf using in vivo confocal imaging to understand defects in the cell body (soma) and processes (*Figure 5A*). We found no significant difference in the soma area at 3 dpf, but at 10 dpf, there is a significant increase in mutant pericyte soma area compared to wild-type (*Figure 5B*). In parallel, wild-type pericytes undergo a slight reduction in soma size from 3 to 10 dpf.

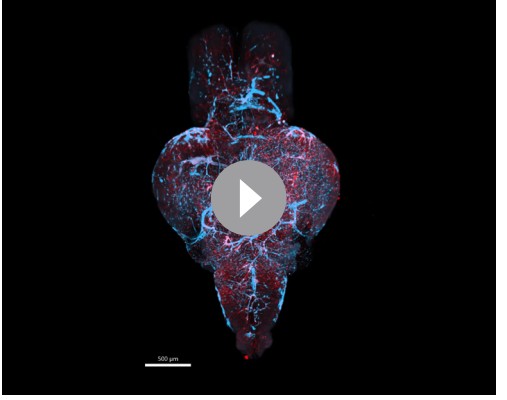

**Video 2.** Rotating view of a cleared *foxf2a* mutant brain at 3 mpf with *pdgfr*β (blue) and *kdrl* (red).
https://elifesciences.org/articles/106720/figures#video2

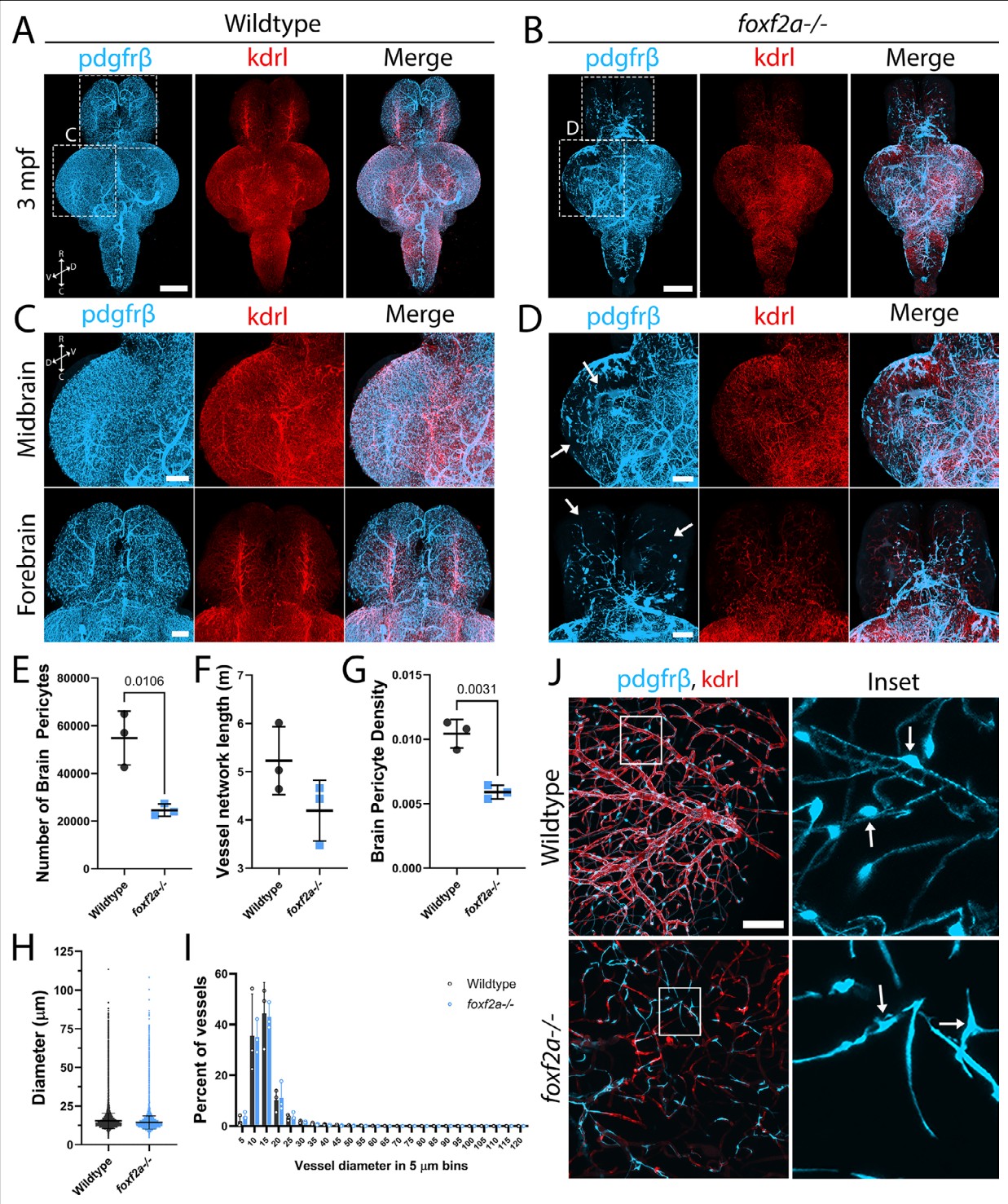

**Figure 3.** *foxf2a* mutants show strong brain vascular defects as adults. (**A–B**) 3D projections of iDISCO-cleared immunostained whole wild-type and *foxf2a⁻/⁻* brains at 3 mpf, viewed ventrally. (**C–D**) Wild-type and *foxf2a* mutant 2 brain regions, viewed dorsally (arrows = defects in coverage). (**E**) Number of brain pericytes in three individual wild-type and mutant brains at 3 mpf detected using Imaris' spot tool and machine learning. (**F**) Total vessel network length in three individual wild-type and mutant brains at 3 mpf using Ilastik and Imaris' filament tool and machine learning. (**G**) Brain pericyte density calculated using number of brain pericytes per meter of vessel length. (**H**) Vessel diameter in three individual wild-type and mutant brains at 3 mpf using Imaris' filament tool and machine learning. (**I**) Percentage of vessel segments in wild-type and mutant brains at 3 mpf segregated by vessel diameter (in 5 µm bins). (**J**) CUBIC-cleared wild-type and *foxf2a* mutant midbrain at 11 mpf (arrows: individual pericyte soma). C=caudal, D=dorsal, *R*=rostral,

*Figure 3 continued on next page*

Figure 3 continued

V=ventral. Statistical analysis was conducted using unpaired t-tests (**E–H**) and ANOVA with Dunnett's post hoc test (**I**). Scale bars, 500 μm (**A–B**), 200 μm (**C–D**), 50 μm (**J**).

The online version of this article includes the following source data and figure supplement(s) for figure 3:

**Source data 1.** All raw quantitative data underlying *Figure 3* and supplements.

**Figure supplement 1.** Workflow of computational analysis of vascular network in adult brains.

**Figure supplement 2.** *foxf2a* mutants show strong brain vascular defects in adulthood.

**Figure supplement 3.** Pericyte heterogeneity in the adult zebrafish brain.

**Figure supplement 4.** *foxf2a* mutants show morphologically unusual *pdgfrβ*-expressing cells and blood vessels in the adult brain.

**Figure supplement 5.** Abnormal blood vessels become apparent in adult *foxf2a* mutant brains.

The low number of brain pericytes in *foxf2a* mutants means that distinct pericyte processes can be distinguished and measured. However, in wild-types, pericyte processes are not easily distinguished. For accurate process length measurements in wild-type, we used multispectral labelling with a pericyte-specific Zebrabow transgenic line (*pdgfrβ:Gal4, UAS:Zebrabow*) (*Pan et al., 2013*; *Whitesell et al., 2019*). Cre mRNA was injected at the one-cell stage to activate random recombination, allowing us to visualize individual neighbouring pericytes (*Figure 5C*). Wild-type processes have a mean length of 81.3 μm vs 102.9 in the mutant at 3 dpf and 96.3 μm vs 147.6 in the mutant at 10 dpf (*Figure 5D*). Thus, the pericyte process lengths are significantly increased in *foxf2a* mutants at 3 and 10 dpf (*Figure 5D*).

Through the study of wild-type processes using multispectral labelling, we observed some differences in the behaviour of adjacent pericyte processes from that published in the adult mouse brain (*Berthiaume et al., 2018a*). While direct contact with no overlap between pericytes is the most common interaction in developing zebrafish, similar to mouse adult brain pericytes (*Figure 5F*), we also see some overlap (*Figure 5E*). The average overlap between wild-type pericyte processes at 10 dpf was 5.9 μm (*Figure 5G*). Together, our data show overlap in brain pericyte processes in wild-type animals.

To further explore mutant pericyte behaviour, we conducted serial imaging during larval development. Over time, some mutant pericyte processes form disconnected bead-like blebs with cell bodies that disappear over time (*Figure 6A*). This pericyte phenotype can occur on vessels with full patency and is not associated with endothelial regression. The bead-like remnants are highly prevalent during larval development in *foxf2a⁻ᐟ⁻* mutants but not present in wild-type (*Figure 6B*). To visualize process degeneration in real-time, we time-lapse imaged from 4 to 5 dpf (*Figure 6C*). The pericyte can undergo a process reminiscent of cell death with soma blebbing and process degeneration phenotypically resembling neural dendrite degeneration or pruning (*Figure 6D*).

In summary, larval *foxf2a* mutant pericytes show reduced numbers, increased soma size, and elongated processes with evidence of process degeneration. In addition, we make the novel observation of overlapping pericyte processes during zebrafish development.

## Foxf2a mutants do not have an impaired capacity to repopulate brain pericytes

Zebrafish have regenerative capacity in various tissues (*Becker et al., 1997*; *Lepilina et al., 2006*; *Otteson and Hitchcock, 2003*), yet *foxf2a* mutant embryos that are pericyte-deficient maintain strong brain pericyte defects through aging, which suggests that either they may not be able to replenish absent/damaged pericytes, or that the smaller size of the initial pool in embryogenesis limits repair such that numbers never can catch up to wild-type. To differentiate these hypotheses, we tested whether *foxf2a* mutants lack the capacity to regenerate pericytes. We reduced pericyte numbers in a *foxf2a* heterozygous incross using a cell ablation strategy. Zebrafish expressing *pdgfrβ:Gal4, UAS:NTR-mCherry,* and *flk:GFP* transgenes were treated with 5 mM metronidazole (MTZ) at 50 hpf for 1 hr which ablates most pericytes. MTZ is a prodrug substrate that elicits cell death in nitroreductase (NTR)-expressing cells due to its cytotoxic derivatives. We then imaged and counted brain pericytes at 3 dpf (a day after treatment) and 10 dpf (recovery).

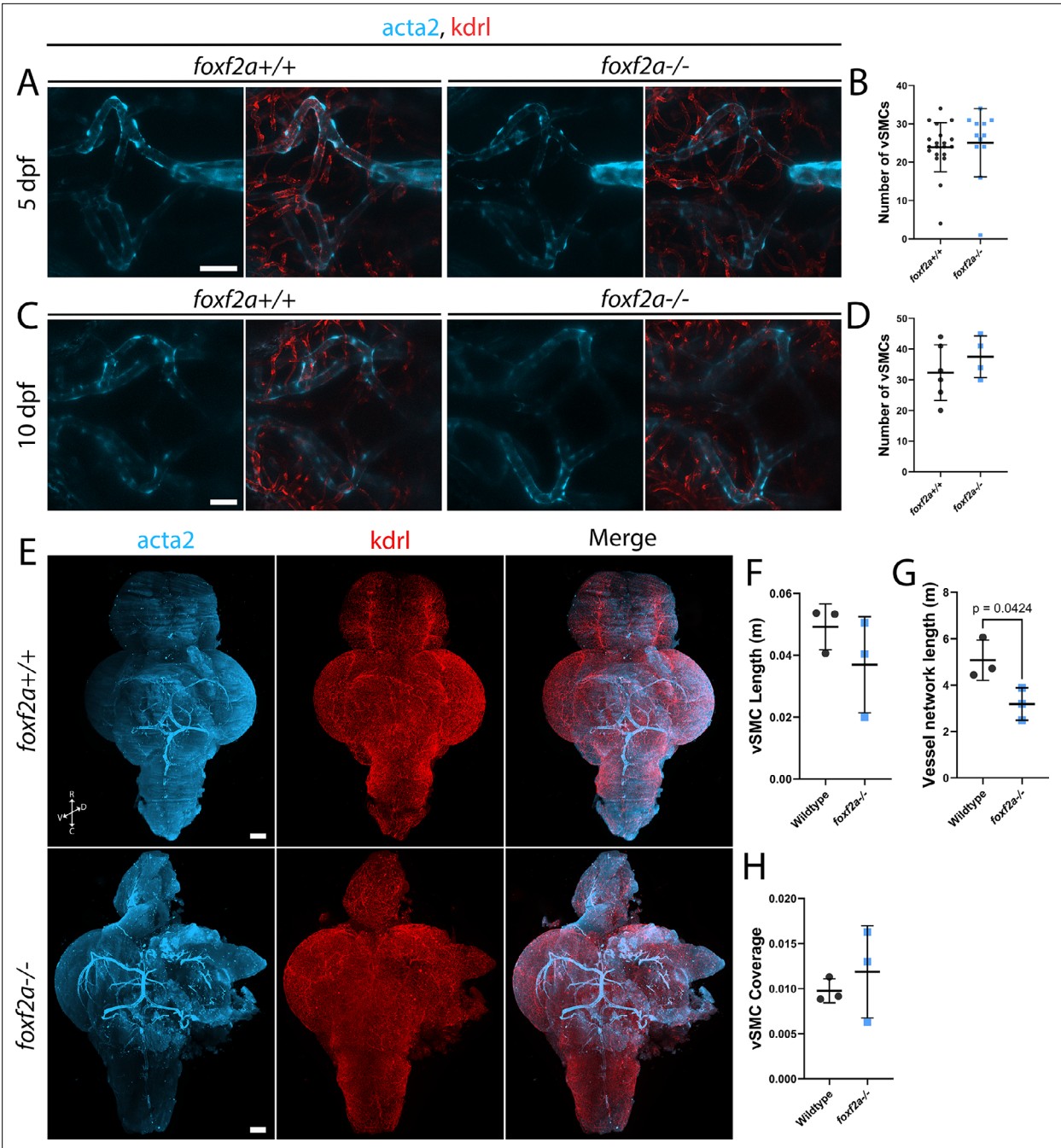

**Figure 4.** Loss of *foxf2a* has no impact on *acta2*-expressing brain vascular smooth muscle cells. (**A**) *foxf2a+/++* and *foxf2a-/-* larvae at 5 dpf showing vSMC coverage in the brain using endothelial (red; *Tg(kdrl:mCherry)*) and vascular smooth muscle cell (vSMC) (light blue; *Tg(acta2:GFP)*) transgenic lines. (**B**) Scatter plot of total vSMCs at 5 dpf. (**C**) *foxf2a+/++* and *foxf2a-/-* larvae at 10 dpf showing vSMC coverage in the brain. (**D**) Scatter plot of total vSMCs at 10 dpf. (**E**) 3D projections of iDISCO-cleared immunostained whole wild-type and *foxf2a⁻/⁻* brains at 6 mpf, viewed ventrally. (**F**) Total vSMC length at 6 mpf using Imaris' filament tool and machine learning. (**G**) Total vessel network length at 6 mpf Ilastic and Imaris' filament tool and machine learning. (**H**) vSMC coverage per total blood vessel network length at 6 mpf. C=caudal, D=dorsal, R=rostral, V=ventral. Statistical analysis was conducted using unpaired t-tests. Scale bars, 20 μm (**A, C**), 200 μm (**E**).

The online version of this article includes the following source data for figure 4:

**Source data 1.** All raw quantitative data underlying *Figure 4*.

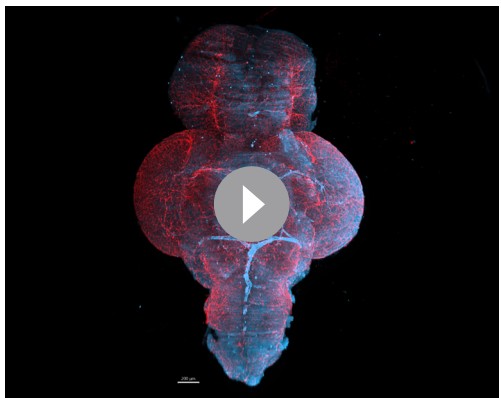

**Video 3.** Rotating view of a cleared wild-type brain at 6 mpf with *acta2* (blue) and *kdrl* (red).
https://elifesciences.org/articles/106720/figures#video3

At 3 dpf, in the vehicle (DMSO) control group, there is the expected significant difference between wild-type and mutant pericyte numbers (*Figure 7A and B*; *Figure 7—source data 1*). When treated with metronidazole, both mutants and wild-types both have a similar, severe reduction in pericytes at 3 dpf post-ablation and are not significantly different from each other (*Figure 7B*). By 10 dpf, pericytes are partially repopulated in both wild-type and mutant ablated groups (*Figure 7C*). Surprisingly, given the initial pericyte number defects in *foxf2a* mutant embryos, we find no significant difference between any groups (*Figure 7D*; mean 70 in wild-type DMSO treated, 53 in mutant DMSO treated, 43 in wild-type MTZ treated, 33 in mutant MTZ treated). This shows that *foxf2a* mutants retain their ability to repopulate pericytes following ablation. This data is important to distinguish the timing and mechanism of pericyte reduction in *foxf2a* mutants. While *foxf2a* mutant pericytes can regenerate following induced catastrophic loss, it cannot compensate for the initially smaller pool of pericytes during embryogenesis. This data suggests that foxf2a plays a critical role in the establishment of a properly sized initial pericyte pool during early development. Regeneration mechanisms, although intact, cannot fully compensate for this initial reduction in pericytes.

## Discussion

Our reduced dosage model of Foxf2 demonstrates disease processes at the cellular level in intact animals, giving insight into pathological changes that occur during CSVD that have not been observed in other models or humans. Our data supports a developmental origin for this type of CSVD, which then progresses and evolves across the lifespan (*Figure 8*).

### Dosage sensitivity of Foxf2

The logic for using a reduced dosage of Foxf2 in our studies is to better match the effect of common population variants leading to CSVD. We previously modelled the effect of a high-risk CSVD and stroke-associated SNV on the expression of FOXF2 in human cells, showing that it can reduce expression by ~50% (*Ryu et al., 2022*). Since zebrafish have two FOXF2 orthologs (foxf2a and foxf2b), foxf2a homozygotes have equivalent FOXF2 dosage as human heterozygotes, assuming that foxf2a and foxf2b not only exhibit similar expression but also share similar functions. Foxf2 is a dosage-sensitive gene, as zebrafish *foxf2a* heterozygotes show a significant reduction in brain pericytes. Similarly, mouse Foxf2 is also dosage sensitive (*Reyahi et al., 2015*). We find that *foxf2a* mutants are variably penetrant, while *foxf2a;foxf2b* double mutants are fully penetrant. Genetic compensation is common with gene duplication (*Kok et al., 2015*; *Rouf et al., 2023*) and would explain variability in disease phenotypes in different individuals with the same mutation.

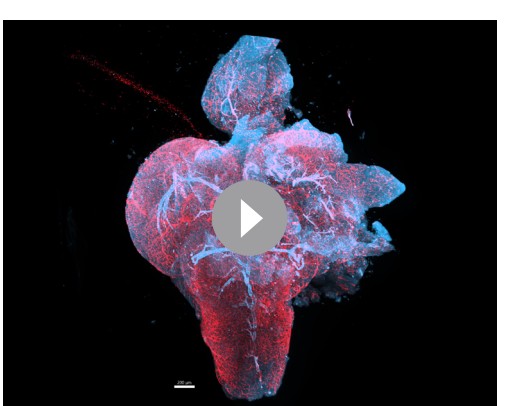

**Video 4.** Rotating view of a cleared *foxf2a* mutant brain at 6 mpf with *acta2* (blue) and *kdrl* (red).
https://elifesciences.org/articles/106720/figures#video4

### Impaired embryonic pericyte coverage in Foxf2 deficiency

We show that numbers of brain pericytes are reduced at multiple developmental and adult stages when *foxf2a* and/or *foxf2a;foxf2b* are lost. The deficiency is significant at every stage,

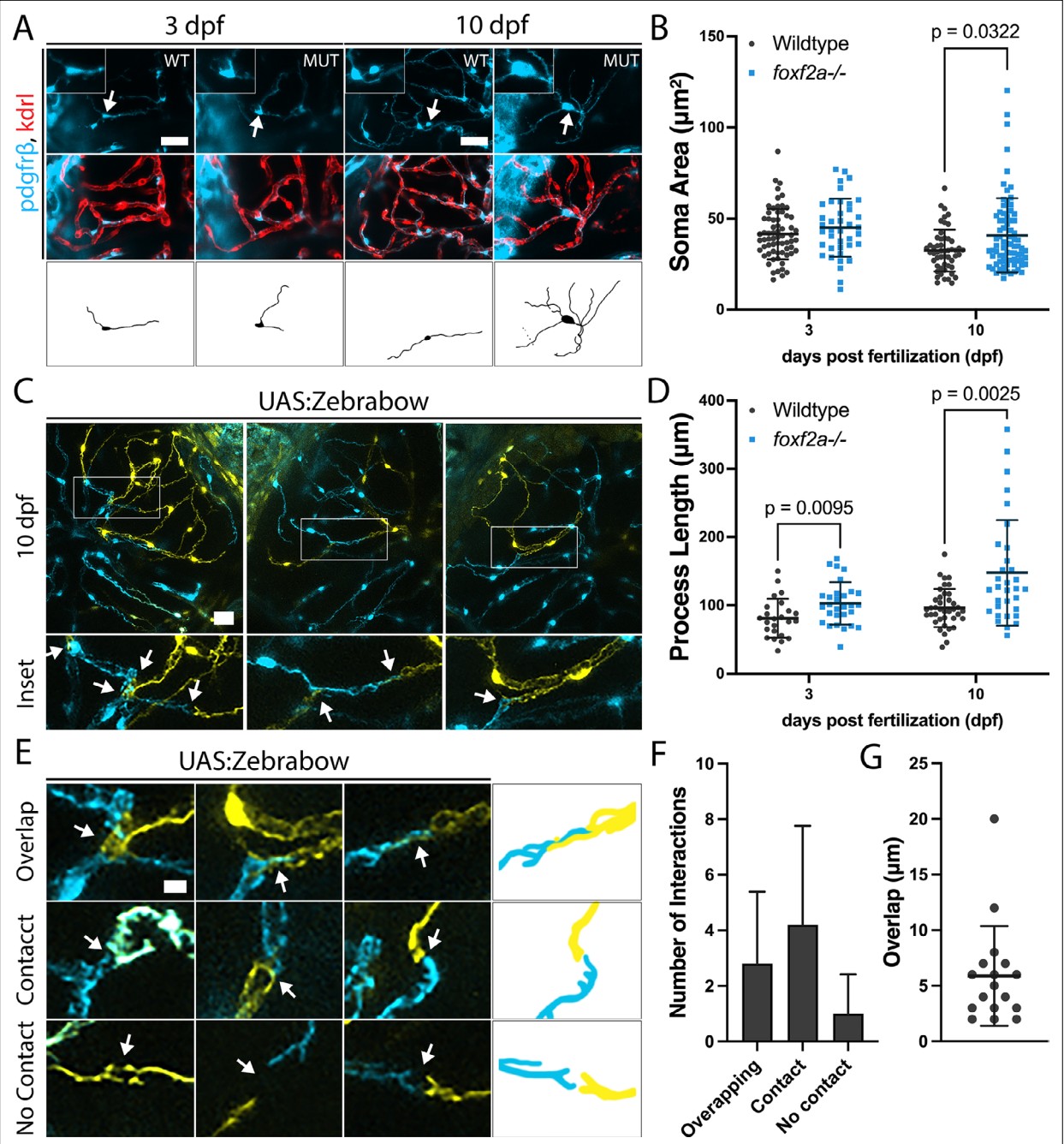

**Figure 5.** *foxf2a* mutant brain pericytes show increased soma size and process length. (**A**) Wild-type and *foxf2a⁻/⁻* mutant brain pericytes at 3 and 10 dpf with tracings of individual pericytes (indicated by arrows). (**B**) Brain pericyte soma area at 3 and 10 dpf. (**C**) Multispectral Zebrabow labelling reveals pericyte-process interactions in the larval brain. (arrows: pericyte interaction points). (**D**) Total process length per pericyte at 3 and 10 dpf. (**E**) Varying pericyte-pericyte interactions at 10 dpf (arrows: interaction points). (**F**) Number of each type of interaction at 10 dpf. (**G**) Length of overlap when process interaction occurs. Statistical analysis was conducted using multiple Mann-Whitney tests in B, a one-way ANOVA with Tukey's test at 3 dpf and a Kruskal-Wallis test with Dunn's multiple comparisons test at 10 dpf in D. Scale bars, 25 μm (**A**), 20 μm (**C**), 5 μm (**E**).

The online version of this article includes the following source data for figure 5:

**Source data 1.** All raw quantitative data underlying *Figure 5*.

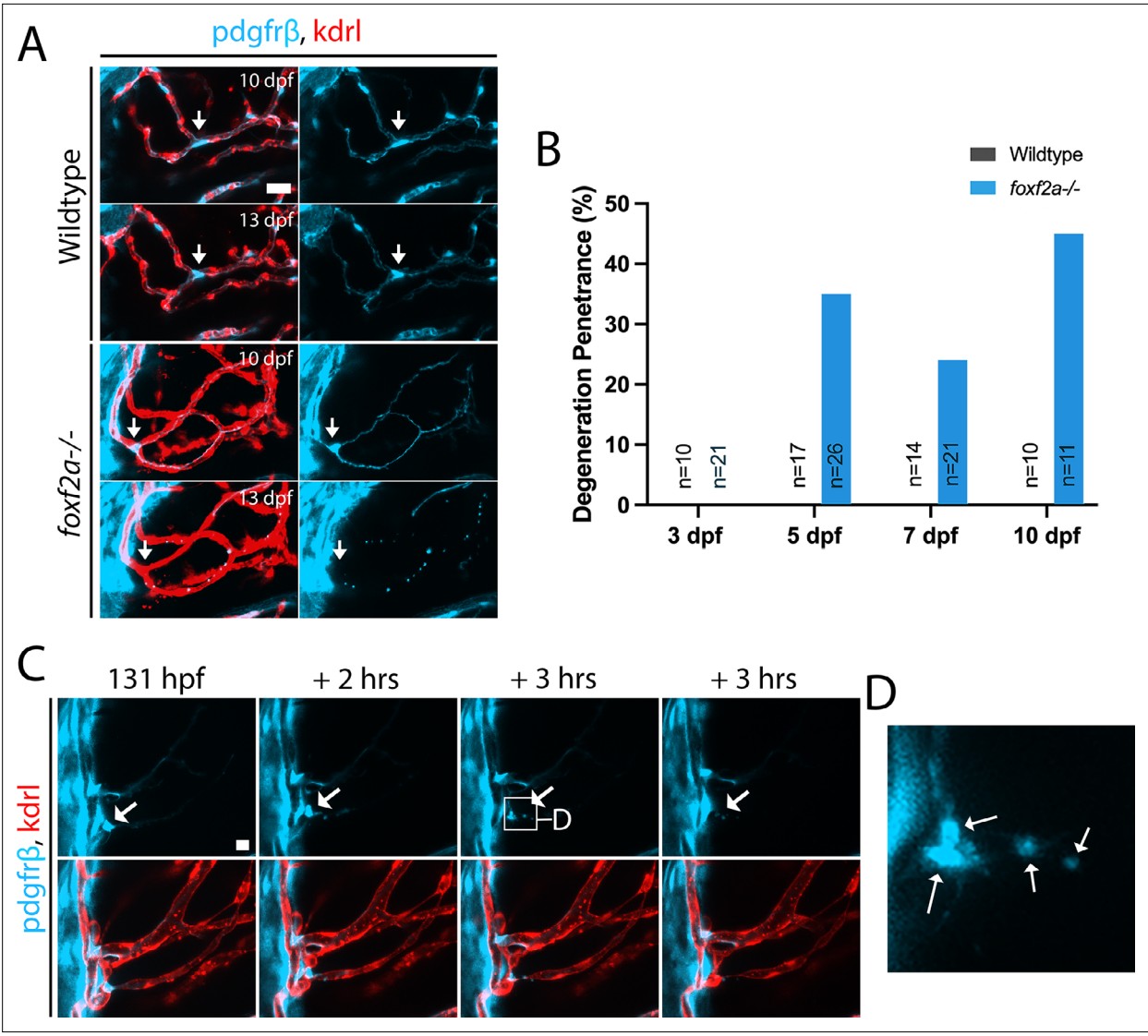

**Figure 6.** *foxf2a* mutant pericytes degenerate. (**A**) *foxf2a*⁻/⁻ mutant pericyte at 10 and 13 dpf with the degenerating process and cell body with a wild-type control from the same brain region (arrows: individual pericyte). (**B**) Bar graph with process blebbing phenotype penetrance in wild-type and mutant brains (n=total samples examined). (**C**) Time-lapse of a *foxf2a*⁻/⁻ mutant midbrain from 4 to 5 dpf (arrows: individual pericyte). (**D**) Inset of mutant pericyte undergoing degeneration (arrows: blebbing). Scale bars, 20 μm (**A, C**).

The online version of this article includes the following source data for figure 6:

**Source data 1.** All raw quantitative data underlying *Figure 6*.

including 3 dpf, the earliest time point that pericytes are robustly observed in development. Deficient numbers could be due to a reduction in the pericyte precursor population (i.e. *nkx3.1* positive cells *Ahuja et al., 2024*), or to an inability of *foxf2a*-deficient cells to differentiate into *pdgfrβ*-positive pericytes. Our data does not allow us to distinguish these, although *pdgfrβ* expression in *foxf2a* mutants is transcriptionally unchanged, suggesting that pericyte number in early development is the primary phenotype. Additional evidence for pericytes being the primary cells affected is that there is no difference in total vessel network length or hindbrain CtA diameter at 3 dpf in *foxf2a* heterozygotes or mutants when pericyte numbers are reduced. A reduction in pericytes is expected to have early consequences, as pericytes provide signals to the endothelium for quiescence and arterial-venous identity (*Mäe et al., 2021* #1750), as well as allowing contractility of cerebral blood vessels in early development (*Bahrami and Childs, 2020*). Fewer pericytes distributed on a normal-sized endothelial network length result in reduced vessel coverage. It is intriguing, therefore, that pericyte

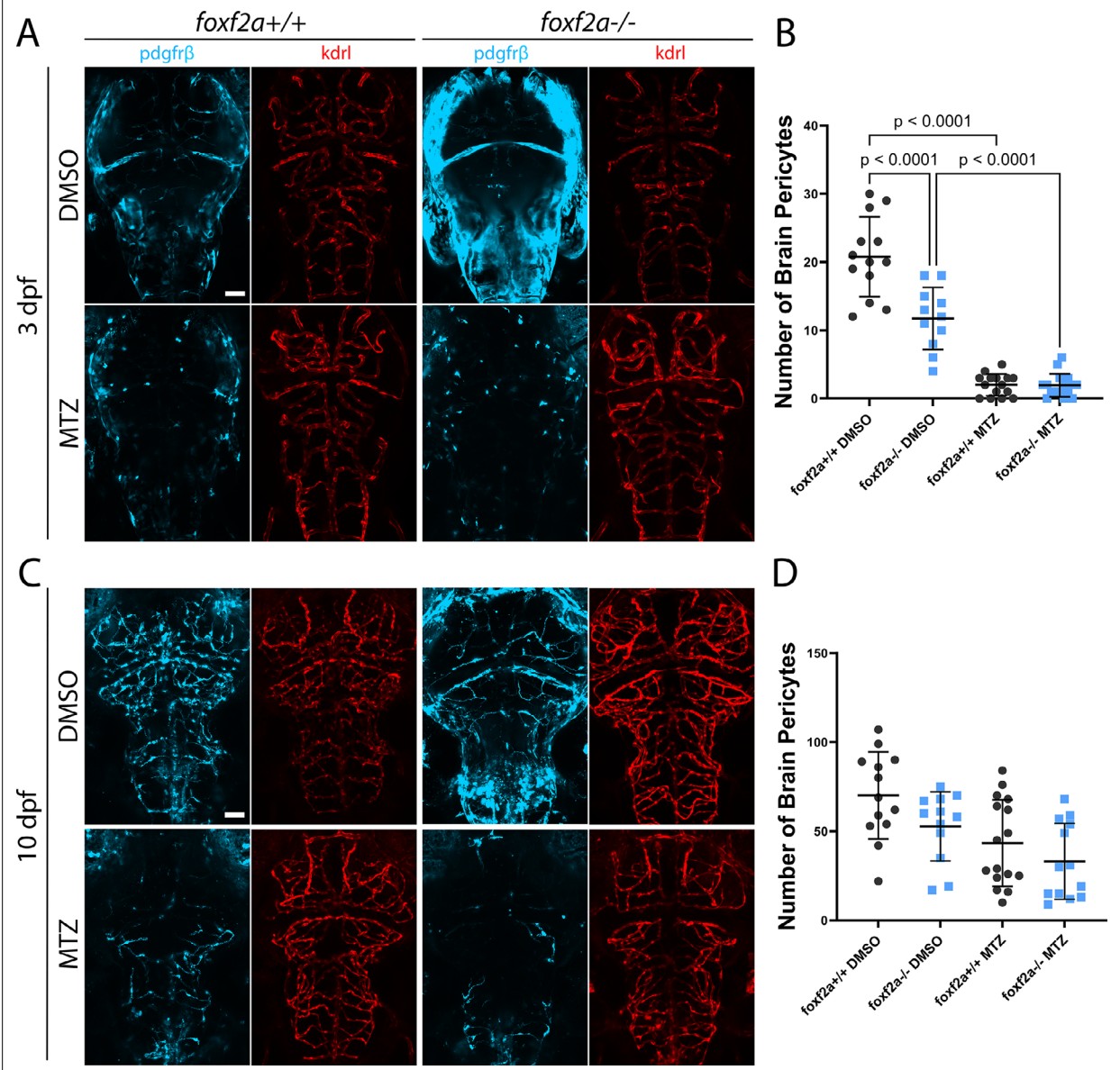

**Figure 7.** *foxf2a* mutants regenerate brain pericytes normally after genetic ablation. Zebrafish brains were imaged using endothelial (red; *Tg(kdrl:GFP)*) and pericyte (light blue; *Tg(pdgfrβ:Gal4, UAS:NTR-mCherry)*) transgenic lines. (**A**) Wild-type and mutant brains at 3 dpf in control (DMSO) and treated (MTZ) groups. (**B**) Total brain pericytes at 3 dpf. (**C**) Wild-type and mutant brains at 10 dpf. (**D**) Total brain pericytes at 10 dpf. Statistical analysis was conducted using a one-way ANOVA (**B**) or Kruskal-Wallis test with Dunn's multiple comparisons test (**D**). Scale bars, 50 μm (**A, C**).

The online version of this article includes the following source data for figure 7:

**Source data 1.** All raw quantitative data underlying *Figure 7*.

process length is increased at both 3 and 10 dpf, potentially to compensate for low pericyte density. A similar elongated pericyte phenotype and reduced coverage is seen in mice with constitutive *Pdgfβ^{ret}* knockout (*Mäe et al., 2021*). The convergent phenotypes after manipulation of two genes (*foxf2a* in fish and *Pdgfβ ^{ret}* in mice) that reduce pericyte number suggest that the intrinsic pericyte response to depletion of pericyte density is to elongate, perhaps to attempt to cover 'naked' vessels. We note that previous mouse *Foxf2* knockouts using a conditional Wnt1-Cre driver saw contrasting results to ours, with increased pericyte numbers, although a similar loss of vascular stability is seen in both models (*Reyahi et al., 2015*). The reason for the difference in the direction in pericyte number may be experimental due to the different knockout technique (conditional in mice which only removes neural

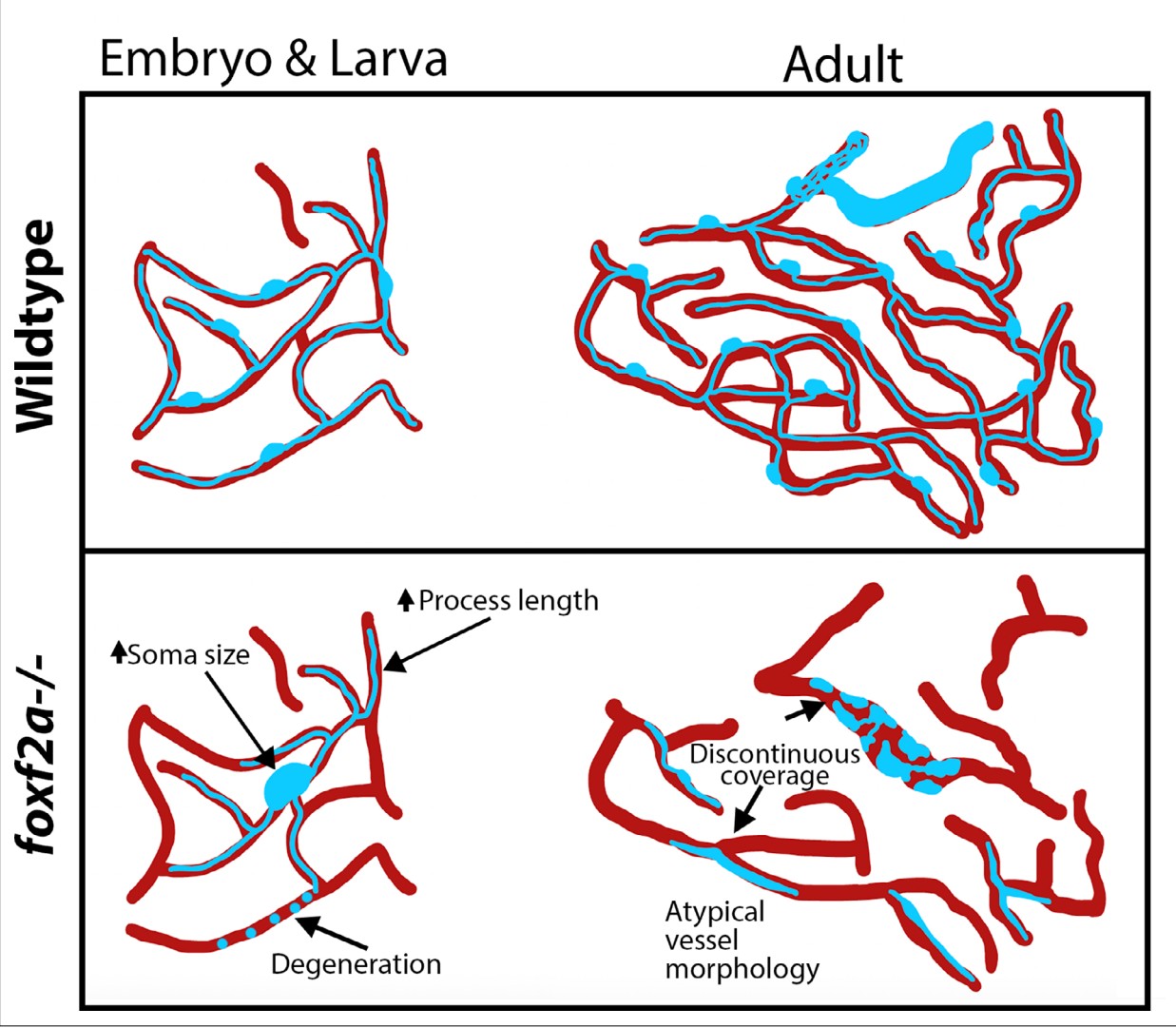

**Figure 8.** Model of *foxf2a* mutant brain pericyte defects over the lifespan. Wild-type pericytes develop normally in the embryo and establish extensive, continuous coverage over vessels by adulthood. *foxf2a* mutant pericytes exhibit abnormal morphology during development that worsens over the lifespan, with mutant vessels developing atypical morphology and discontinuous coverage.

crest-derived pericytes vs. full knockout which removes mesodermal, neural crest-derived and endothelial *foxf2a* in fish), or the difference could reflect a species-specific difference.

### Insights on fundamental pericyte process properties

The reasons behind the hypergrowth of pericyte processes in *foxf2a*$^{-/-}$ mutant brains is unclear. Berthiaume et al. (*Berthiaume et al., 2018b*) propose that repulsive interactions among pericytes in the adult brain establish boundaries between adjacent pericytes, preventing their processes from overlapping. In this case, one might hypothesize that *foxf2a* mutant pericytes, which are less dense along vessels, would lack feedback from adjacent pericytes, leading to uncontrolled growth.

In mice, adult brain pericytes form a non-overlapping network along capillaries, although their processes occasionally approach each other without overlapping (*Berthiaume et al., 2018a*; *Hill et al., 2015*). Overlap between pericyte processes have not been well studied in the developing brain, but *pdgfrb*$^{+ve}$ cells in developing zebrafish fin also overlap (*Leonard et al., 2022*). Strikingly, using multispectral (Zebrabow) labeling, we find processes overlap in the developing zebrafish brain. Overlaps occur on capillary segments and at branch points. Previous studies in mice were conducted in the adult brain, and it is possible that overlapping processes in the developing brain are transient, zebrafish-specific, or were potentially overlooked in the mouse experiments.

We have also observed forked extensions at the tips of some pericyte projections at 10 dpf in wild-type embryos, where pericytes are not in contact. Whether these unique structures are involved in attractive/repulsive signals between pericytes or help determine the direction and extent of process growth is unclear, but understanding the fundamental mechanisms governing pericyte-pericyte process interactions would yield valuable insights into development and disease, as well as into how pericyte depletion results in abnormally long processes.

## Morphological abnormalities in pericyte soma in foxf2a mutants

A second morphological abnormality of *foxf2a*-deficient pericytes is that their soma size is increased at 10 dpf, which may also be a compensatory change. Pathological changes in soma size have not been observed in brain pericytes. However, increased neuronal soma size is observed in Amyotrophic Lateral Sclerosis or Lhermitte-Duclos (*Dukkipati et al., 2018*; *Kwon et al., 2001*) and is linked to mTor signaling (*Kwon et al., 2001*). In neuronal populations, moderate soma swelling can be an adaptation for survival, while rapid, drastic swelling indicates imminent death (*Rousseau et al., 1999*). It is, therefore, not surprising that we observe degeneration of pre-existing pericytes in *foxf2a* mutant but not wild-type animals. This degeneration phenotypically resembles neuronal dendrite remodelling and pruning during development (*Fukui et al., 2012*; *Greenwood et al., 2007*). Further work to test whether pericytes share similar mechanisms of degeneration in response to stress or cellular damage may provide further insight into phenotypic progression.

## Changes in foxf2a mutants across the lifespan

For the first time, we have examined how vascular phenotypes change in the adult brain using iDISCO clearing. We find that adult soma take on very unusual, 'stiff' shapes and become almost indistinguishable from processes. The adult processes are shorter and more linear than the embryonic processes. Hypertrophic embryonic pericytes either do not survive to adulthood or undergo morphological changes over time. Given the limited knowledge in this area, the underlying factors driving this shift in pericyte morphology are unclear. Actin is present near the plasma membrane of cells, where it provides structural and mechanical support, enabling motility and determining cell structure. Given the abnormal shapes of adult pericytes, alterations in actin may contribute to their irregular morphology.

In the adult *foxf2a* mutant brain, abnormal *pdgfrβ*-expressing cells are observed on large-calibre vessels and likely represent vSMCs. We observed large-calibre aneurysm-like vessels, suggesting that vSMCs may lose functionality over time, thereby increasing the vulnerability of underlying vessels to dilation and weakening. Loss of *acta2* and, subsequently, vSMCs has been associated with conditions, such as thoracic aortic aneurysms and dissections (*Guo et al., 2009*). However, we found no significant change in either vSMC cell number in embryonic development or in network length of vSMCs in the adult brain, suggesting that the primary phenotype in *foxf2a* mutants is in pericyte cells and that any changes to vSMCs are likely secondary.

Similarly, brain vessel diameter and network length are not significantly altered in embryonic *foxf2* mutants, but in 6-month-old adults, the vessel network length is significantly decreased. Our data suggest that *foxf2* deficiency contributes to cumulative vascular defects. Adult endothelial defects may be secondary to pericyte defects, although *foxf2a* is expressed in both pericytes and endothelial cells, and both cell types may be affected autonomously.

## The initial pericyte pool is critical for lifelong vascular health

To understand the mechanism by which foxf2a influences pericyte numbers, we needed to distinguish between its role in early development and later roles. Our lab recently demonstrated that pericyte precursors express *nkx3.1* and *foxf2a* before pericyte differentiation and prior to expression of canonical markers, such as *pdgfrβ* (*Ahuja et al., 2024*). Here, we show that the pericyte pool in *foxf2a* mutants is reduced at the earliest embryonic stage that it can be measured. Zebrafish are remarkably regenerative and able to repair cardiac, retinal, spinal cord, and pancreatic damage among other tissues (*Poss and Tanaka, 2024*). We, therefore, anticipated that *foxf2a* mutants might be able to regenerate new pericytes to replace lost pericytes. This was tested using genetic ablation of pericytes. Ablation is very efficient in mutants and wild-type. Surprisingly, 7 days after ablation, *foxf2a* mutants show recovery of pericyte numbers. Thus, when placed under extreme stress, embryonic *foxf2a*

mutants can regenerate pericytes; yet, under baseline conditions, *foxf2a* mutant pericytes do not replenish. This experiment is important to distinguish an underlying mechanism of the *foxf2a* pericyte phenotype. Our data suggest that the earliest and most important difference in *foxf2a* mutant pericytes is their low initial number. Even though some repair is possible, it is not enough to compensate. Reduced pericyte density is associated with elongated processes, enlarged soma, and degeneration in mutants, which are likely signs of cellular stress. Over a lifetime, the deficiency leads to progressive damage to the vascular network. Determining that there is a critical embryonic window for developing a robust pericyte precursor pool helps us focus on early interventions to mitigate these deficits before damage over the lifespan worsens. The necessity of Foxf2 during development for the establishment of proper brain vasculature was also seen in a mouse Foxf2 knockout (*Reyahi et al., 2015*). Further research into the early role of *foxf2a* in establishing initial pericyte specification/differentiation, and the impact of early vascular defects on disease progression, will be crucial for developing strategies to prevent and treat cerebrovascular conditions.

## Materials and methods
### Zebrafish husbandry and strains

All procedures were conducted in compliance with the Canadian Council on Animal Care, and ethics approval was granted by the University of Calgary Animal Care Committee (AC21-0169). Embryos were maintained in E3 medium (5 mM NaCl, 0.17 mM KCl, 0.33 mM CaCl2, 0.33 mM MgSO4, pH = 7.2) (*Westerfield, 1995*) at 28 °C. Larvae up to 10 dpf were maintained in an incubator with a light cycle (14 hr light, 10 hr darkness), with daily water changes and feeding. As zebrafish sex is not determined until 28 dpf, sex is not considered in embryo and larval experiments. For adult experiments, results were compared by sex where sufficient n's were available. Results from the progeny of het in-crosses were blinded in that they were quantified before genotyping. All reagents and strains are listed in Key resources table. All experiments included wild-type as a comparison group, and developmental stages, n's, and genotypes are noted for each experiment source data table for each figure.

**Key resources table**

| Reagent type (species) or resource | Designation | Source or reference | Identifiers | Additional information |
|---|---|---|---|---|
| Strain, strain background (*D. rerio*) | foxf2a[ca71] | *Ryu et al., 2022* | ZDB-ALT-230214–10 | |
| Strain, strain background (*D. rerio*) | foxf2b[ca22] | *Chauhan et al., 2016* | ZDB-ALT-160809–1 | |
| Strain, strain background (*D. rerio*) | Tg(acta2:GFP)[ca7] | *Whitesell et al., 2014* | ZDB-ALT-120508–1 | |
| Strain, strain background (*D. rerio*) | Tg(4xUAS:Zebrabow-B)[a133] | *Pan et al., 2013* | ZFIN: ZDB-ALT-130816–3 | |
| Strain, strain background (*D. rerio*) | Tg(flk:GFP)[la116] | *Choi et al., 2007* | ZDB-TGCONSTRCT-070529–1 | |
| Strain, strain background (*D. rerio*) | Tg(kdrl:mCherry)[ci5] | *Proulx et al., 2010* | ZFIN: ZDB-ALT-110131–57 | |
| Strain, strain background (*D. rerio*) | Tg(UAS:NTR-mCherry)[c264] | *Davison et al., 2007* | ZDB-ALT-070316–1 | |
| Strain, strain background (*D. rerio*) | TgBAC(4xUAS:EGFP)[mpn100Tg] | *DeMaria et al., 2013* | ZFIN: ZDB-TGCONSTRCT-140812–1 | |
| Strain, strain background (*D. rerio*) | TgBAC(pdgfrβ:GAL4FF)[ca42] | *Whitesell et al., 2019* | ZFIN: ZDB-ALT-200102–2 | |
| Strain, strain background (*D. rerio*) | TgBAC(pdgfrβ:EGFP)[ca41Tg] | *Whitesell et al., 2019* | ZDB-TGCONSTRCT-160609–1 | |
| Antibody | anti mCherry, rat monoclonal | Thermo Fisher, M11217 | RRID:AB_2536611 | 1 in 500 |
| Antibody | anti Green Fluorescent Protein, mouse monoclonal | Clontech, 3 P 632380 | RRID:AB_10013427 | 1 in 500 |

*Continued on next page*

*Continued*

| Reagent type (species) or resource | Designation | Source or reference | Identifiers | Additional information |
|---|---|---|---|---|
| Antibody | Donkey anti mouse 488 | Thermo Fisher, A-21202 | RRID:AB_141607 | 1 in 500 |
| Antibody | Goat anti rat 555 | Thermo Fisher, A-21434 | RRID:AB_2535855 | 1 in 500 |
| Commercial assay or kit | Fluoromount-G Mounting Medium with DAPI | Invitrogen | E141201 | |
| Commercial assay or kit | Fluoromount-G Mounting Medium | Thermo Fisher | 00-4958-02 | |
| Commercial assay or kit | Taqman SNP genotyping kit for foxf2b$^{ca22}$ | Applied Biosystems | ANAACEC | |
| Commercial assay or kit | KAPA2G Fast Hotstart Genotyping Mix | Roche | KK5621 | |
| Chemical compound, drug | Dimethylsulfoxide | Sigma | D8418 | |
| Chemical compound, drug | Phenylthiourea | Sigma | P7629 | |
| Chemical compound, drug | UltraPure Agarose | Invitrogen | 16520–050 | |
| Chemical compound, drug | Metronidazole | Sigma | M3761 | |
| Software, algorithm | Imaris 10.3 | Oxford Instruments | RRID:SCR_007370 | |
| Software, algorithm | Ilastic | https://www.ilastik.org/ | RRID:SCR_015246 | |
| software, algorithm | Fiji (ImageJ) | *Schindelin et al., 2012* | RRID:SCR_002285 | |
| Software, algorithm | GraphPad Prism 10 | Graphpad | RRID:SCR_002798 | |
| Software, algorithm | Adobe Photoshop | Adobe | RRID:SCR_014199 | |
| Software, algorithm | VesselMetrics | *McGarry et al., 2024* Microvasc Res, 2024 | | |
| Commercial assay or kit | HCR Probe (v3.0) ndu4al2a | Molecular Instruments | | |
| Commercial assay or kit | HCR Probe (v3.0) kdrla | Molecular Instruments | | |
| Commercial assay or kit | HCR Probe (v3.0) pdgfrβ | Molecular Instruments | | |
| Commercial assay or kit | HCR Probe (v3.0) foxf2a | Molecular Instruments | | |
| Commercial assay or kit | HCR Probe (v3.0) foxf2b | Molecular Instruments | | |
| Sequence-based reagent | foxf2a-genotyping-forward | IDT | ATG CAC TCG GCT CTC CAA AA | |
| Sequence-based reagent | foxf2a-genotyping-reverse | IDT | GAT CGC CAT GAC TAT CGG GG | |

## Genotyping

Adult fish were anesthetized in 0.4% Tricaine (Sigma) and placed on a cutting surface. A small portion of the tip of the fin was clipped using a razor blade, and the fish were returned to the system to recover. For developmental DNA isolation, whole embryos, or larvae (up to 10 dpf) were anesthetized in 0.4% Tricaine prior to sampling.

Genomic DNA (gDNA) was extracted using the HotSHOT DNA isolation protocol, adapted from *Meeker et al., 2007*. To isolate gDNA, tissue or embryo was placed in 50 µL of Base Solution and incubated at 95 °C for 30 min. 5 µL of Neutralization Solution was added to neutralize the reaction. Samples were spun down in a mini centrifuge, and DNA was sampled from the top portion of the solution to avoid undigested samples.

Zebrafish were genotyped for target genes or the presence of transgenes using the KAPA2G Fast HotStart Genotyping Mix as per the manufacturer's instructions. *foxf2* mutants were genotyped using *foxf2a* primers (Key resources table). A custom TaqMan probe for *foxf2b$^{ca22}$* (ANAACEC) was obtained from Applied Biosystems. Samples were genotyped using the QuantStudio 6 Flex Real-Time PCR

System (Applied Biosystems) with the wild-type allele reported in FAM and the mutant allele reported in VIC.

## In situ hybridization

Custom probes for *foxf2a* (PRD069), *nduf4al2* (RTD146), *kdrl* (PRI089), *pdgfrβ* (PRA654), and *foxf2b* (PRF462) were obtained from Molecular Instruments (Los Angeles, CA) for Hybridization Chain Reaction (HCR) in situ hybridization. In situ staining was performed according to the manufacturer's instructions. Samples were permeabilized with proteinase K (1 mg/mL stock; Invitrogen, 4333793) at various times and concentrations, depending on their age.

## Integrated intensity

*pdgfrβ* and *foxf2b* intensity was obtained using ImageJ from in-situ stained embryos at 3 dpf. Mean fluorescent intensity was measured using the freehand selection tool to circle each pericyte. The background mean intensity was subtracted from the pericyte mean intensity to standardize each measurement, and the intensities of each fish were averaged. The same protocol was used to measure the intensity of the *pdgfrβ* transgene in 3 dpf live embryos.

## Total RNA extraction and cDNA synthesis

Total RNA was extracted using the Research RNA Clean & Concentrator-5 Kit (Zymo Research, R1015) with some modifications. 50 dechorionated embryos were collected at 48 hpf and homogenized in TRIZOL (Ambion, 15596026) reagent with a syringe needle. Briefly, samples were centrifuged at 16,000 rpm for 2 min to remove pigment. After separating the solution from the precipitate, chloroform was added, and the mixture was centrifuged at 12,000 rpm for 15 min at 4 °C. The aqueous layer containing nucleic acids was removed, and an equal volume of 95–100% ethanol was added, then mixed well. The rest of the protocol followed the manufacturer's instructions using the Zymo-spin IC column. After eluting RNA with RNase/DNase-free water, samples were quantified using a Nanodrop to evaluate RNA quality. For cDNA synthesis, qSCRIPT cDNA Supermix (QuantaBio, 95048–100) was used according to the manufacturer's instructions and stored at –20 °C.

## RT-qPCR

To assess relative gene expression, real-time quantitative PCR (RT-qPCR) was carried out using custom-designed primers. PowerUP SYBR Green Master Mix (Applied Biosystems, A25742) was utilized on a QuantStudio 6 Flex Real-Time PCR System (Applied Biosystems). Cycle conditions followed the protocol of a PowerUP SYBR Green standard reaction: 50 °C for 2 min, 95 °C for 2 min, and 40 cycles of 95 °C for 15 seconds and 60 °C for 60 s. To calculate the fold change in gene expression, mutants were compared to wild-type using ΔΔCT calculations (*Livak and Schmittgen, 2001*).

## Brain dissection

Zebrafish were first euthanized in Tricaine and mounted on a Sylgard gel plate with dissection pins to stabilize them. Scissors were used to sever the brain stem entirely at the base of the head, posterior to the hindbrain. The eyes were removed with forceps/micro-scissors, and the optic stalks were cut. An incision was then made at the posterior portion of the skull plate anteriorly. Forceps were used to pull back the skull plate and orbital bones. Any nerve connections were severed, and the brains were removed and immediately fixed in ice-cold 4% PFA overnight at 4 °C.

## Brain tissue clearing

Whole zebrafish brains were cleared by either CUBIC (*Susaki et al., 2015*), or a modified iDISCO+ (*Renier et al., 2016*) protocol. For CUBIC, samples were incubated in a 1:1 ratio of Reagent 1 (25% Quadrol (Sigma, 122262), 25% Urea, and 15% Triton X-100):H$_2$O overnight at 37 °C. Next, brains were incubated in 100% Reagent 1 at 37 °C until the tissue was transparent. Finally, brains were rinsed in PBS, mounted in 2% low melting point agarose and re-placed in 100% Reagent 1 at 37 °C until the tissue was transparent.

For iDISCO, the samples were first dehydrated in methanol:H$_2$O dilution series for 30 min each, then chilled at 4 °C. Next, samples were incubated overnight in a 1:3 ratio of dichloromethane (DCM; Sigma, 270997):methanol at room temperature. The following day, samples were washed in methanol

and then chilled at 4 °C before bleaching in fresh 5% $H_2O_2$ in methanol overnight at 4 °C. Then, samples were rehydrated in methanol:H2O dilution series for 30 min each, then washed in PTx.2 (0.2% Triton X-100 in PBS) twice over 2 hr at room temperature. Next, samples were incubated in permeabilization solution (0.3 M glycine, 20% DMSO in 400 mL PTx.2) at 37 °C for up to two days, after which the samples were rinsed in PBS twice over 1 hr. Samples were then washed three times over 2 hr in 0.5 mM SDS/PBS at 37 °C for three days before being incubated in primary antibody (Key resources table) in 0.5 mM SDS/PBS at 37 °C for two more days. The primary antibodies were refreshed in PTx.2 and left for another 4 days before overnight washing in PTwH (10 µg/ml heparin and 0.2% Tween-20 in PBS) at 37 °C. Samples were left in secondary antibodies (Key resources table) in PTwH/3% NSS at 37 °C for 3 days, refreshed and left for another 3 days before washing overnight in PTwH. After immunolabeling, brains were mounted in 2% low melting agarose (LMA) and dehydrated in methanol:$H_2O$ dilution series for 30 min each and left overnight at room temperature. Next, brains were incubated in a 1:3 ratio of DCM: methanol at room temperature for 3 hr before washing in 100% DCM for 15 min twice. Finally, brains were incubated in ethyl cinnamate (Sigma, NSC6773) for 2 hr before replacing the solution and incubating overnight at room temperature.

## Tissue sectioning

For vibratome sections, brains were fixed in 4% PFA, washed, and mounted in 4% LMA (Invitrogen, 16520–050) before sectioning with a vibratome (Leica, VT1000S) to obtain a series of transverse sections at 50 µm.

## Cre mRNA injections

Wild-type fish on a Zebrabow transgenic background (*Pan et al., 2013*) were injected with 1 pg of Cre mRNA at the one-cell stage. Embryos were kept in E3 at 28 °C until imaging.

## Immunofluorescence

Sections for immunofluorescence were briefly washed in PBS, followed by 2% Triton X-100 in PBS for permeabilization. Sections were moved from the permeabilization solution into blocking buffer (5% goat serum, 3% BSA, 0.2% Triton X-100 in PBS) for 1 hr before incubation in primary antibody. Sections were then washed and left in secondary antibodies for 3 hr at room temperature. After incubation, sections were washed, mounted, and cover-slipped with Fluoromount-G Mounting Medium, with DAPI or without counterstain.

## Drug treatments

Embryos were dechorionated prior to treatment, and all drug treatments included DMSO at an equivalent concentration to the drug solution as a control (Sigma, D8418). Treatments were performed in a 24-well plate with approximately 15 embryos per well. For pericyte ablation experiments, 5 mM Metronidazole (MTZ; Sigma, M3761) was applied at 50 hpf for 1 hr with a DMSO control.

## Microscopy

Imaging fluorescent transgene expression in live embryos, antibody staining, and fluorescent staining were completed using an inverted laser scanning confocal microscope (LSM900; Zeiss) with a 10 X (0.25 NA), 20 X (0.8 NA), 40 X water (1.1 NA), or 60 X (1.4 NA) oil objectives. Laser wavelengths included blue (405 nm), green (488 nm), red (561 nm), and far red (640 nm). Embryos were maintained in phenylthiourea (PTU) from 24 hpf onwards to prevent pigment development, anesthetized in 0.4% tricaine and mounted in 0.8% LMA dissolved in E3, on a clear imaging dish. In some cases, live samples were retrieved from the agarose following imaging for further imaging at later time points, genotyping, or other data collection. Images were processed using Zen Blue and Fiji (*Schindelin et al., 2012*) software.

Imaging of whole adult brains was completed using the Light-sheet microscope (Ultramicroscope II equipped with a SuperPlan module; Miltenyi Biotec) with a 4 X (0.35 NA) objective at 1.0 zoom. Laser wavelengths included green (488 nm) and red (561 nm). Images were processed using Zen Blue, Fiji, and Imaris software.

## Vessel network quantitation

Confocal images from embryos were analyzed using the Python-based software tool Vessel Metrics (*McGarry et al., 2024*). The total vessel network length was measured starting from below the middle cerebral vein and dorsal longitudinal vein until the emergence of the basal communicating artery. Vessels included in measurements are as follows: middle mesencephalic central arteries, posterior mesencephalic central arteries, the primordial hindbrain channels, and the posterior region of the basilar artery. The forebrain vessels (i.e. anterior cerebral veins) were not included. Blood vessel diameter was restricted to the horizontal hindbrain central arteries, which were comparable between all images.

Adult vessel network length was measured using Ilastik (*Berg et al., 2019*) and Imaris (version 10.2.0; Bitplane AG, Oxford Instruments) software. Vessels were annotated in Ilastik to obtain a probability map. The probability map was imported into Imaris as a new channel for each image. The surface tool with machine learning was then used to annotate the vessels a second time, using the Ilastik map as the training guide. A mask was created from this surface. The filament tool was then used to create a 3D network of the vessel, using the surface mask as a training guide. The total network length and the diameter of each vessel segment were exported from the filament statistics tab.

Smooth muscle cell coverage was measured using the same pipeline; however, the raw channel was used for the surface tool annotation without using Ilastik. Total network length was exported from the filament statistics tab.

## Pericyte quantitation

Embryonic pericytes were manually counted using the Fiji counting or tracing tool on flattened Z-stacks from confocal images of the whole zebrafish head. Pericyte counting and analysis were restricted to the mid and hindbrain regions for all metrics. For pericyte process lengths, individual processes extending from a single soma were measured and summed to determine the total process length per pericyte (µm). In instances where two processes appeared to merge or cover the same blood vessel, half of the total length was added to each pericyte. For Zebrabow images, only pericytes with processes distinguishable from neighbouring pericytes were measured. For soma size, individual cell bodies were traced, and the area was determined by the software in $µm^2$.

Adult pericytes were counted using the spot tool in Imaris. Using the raw channel, the spot tool was trained to identify pericyte cell bodies. The total spot number, equivalent to the number of pericyte cell bodies, was exported from the spot statistics tab. A mask of the spot tool was created to better visualize pericytes.

## Statistics

All statistical analyses were performed using GraphPad Prism 10, with significance determined by $p < 0.05$. Statistical tests conducted are included in figure captions. If no significance is indicated, it is not significant. The D'Agostino-Pearson test was used to assess the normality of data sets, and in cases where the data did not pass the normality test, non-parametric statistics were used. Experimental N's are reported in source data accompanying each figure. Results are expressed on graphs as mean ± standard deviation (SD). Only significant p-values are indicated.

## Materials availability

All materials used in this study are freely available on request.

## Acknowledgements

This work was funded by the Canadian Institutes of Health Research PJT-183631. MFG received a Canada Graduate Scholarship Master's from the Canadian Institutes of Health Research, the Alberta Graduate Excellence Scholarship (AGES) for Master's Research from the Province of Alberta, and a Biochemistry and Molecular Biology Department scholarship from the University of Calgary. EH received the Alvin Libin Graduate Scholarship in Cardiovascular Research, the Alberta Children's Hospital Research Institute (ACHRI) Graduate Scholarship for Master's Research, the ACHRI Graduate Scholarship for Doctoral Research, and the Alberta Graduate Excellence Scholarship (AGES) for Doctoral Research from the Province of Alberta. We acknowledge the Alberta Children's Hospital

Research Institute and Hotchkiss Brain Institute Imaging facilities for microscopes and technical support.

## Additional information

### Funding

| Funder | Grant reference number | Author |
|---|---|---|
| Canadian Institutes of Health Research | PJT-183631 | Sarah J Childs |

The funders had no role in study design, data collection and interpretation, or the decision to submit the work for publication.

### Author contributions

Merry Faye E Graff, Conceptualization, Formal analysis, Validation, Investigation, Visualization, Methodology, Writing – original draft, Writing – review and editing; Emma EM Heeg, Conceptualization, Formal analysis, Investigation, Methodology, Writing – review and editing; David A Elliott, Resources, Software, Methodology; Sarah J Childs, Conceptualization, Resources, Supervision, Writing – original draft, Project administration, Writing – review and editing

### Author ORCIDs

Merry Faye E Graff ⓘ https://orcid.org/0009-0006-6861-5759
Emma EM Heeg ⓘ https://orcid.org/0009-0002-0879-133X
Sarah J Childs ⓘ https://orcid.org/0000-0003-2261-580X

### Ethics

This study was performed in strict accordance with the recommendations CCAC guidelines: Zebrafish and other small, warm-water laboratory fish from the Canadian Council on Animal Care. All animals were handled according to the approved institutional University of Calgary Animal Care Committee (AC21-0169).

Reviewer #1 (Public review): https://doi.org/10.7554/eLife.106720.3.sa1
Reviewer #2 (Public review): https://doi.org/10.7554/eLife.106720.3.sa2
Reviewer #3 (Public review): https://doi.org/10.7554/eLife.106720.3.sa3
Author response https://doi.org/10.7554/eLife.106720.3.sa4

## Additional files

### Supplementary files

MDAR checklist

### Data availability

Numerical data for all experiments is included for each figure.

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
