## [Editor Report · eLife Assessment]

This study provides **important** insights into mural cell dynamics and vascular pathology using a zebrafish model of cerebral small vessel disease. The authors present **convincing** evidence that partial loss of foxf2 function results in progressive, cell-autonomous defects in pericytes accompanied by endothelial abnormalities across the lifespan. By leveraging advanced in vivo imaging and genetic approaches, the work establishes zebrafish as a powerful and relevant model for dissecting the cellular mechanisms underlying cerebral small vessel disease.

---

## [Referee Report · Reviewer #1 (Public review)]

Summary:

The paper by Graff et al. investigates the function of foxf2 in zebrafish to understand the progression of cerebral small vessel disease. The authors use a partial loss of foxf2 (zebrafish possess two foxf2 genes, foxf2a and foxf2b, and the authors mainly analyze homozygous mutants in foxf2a) to investigate the role of foxf2 signaling in regulating pericyte biology. The find that the number of pericytes is reduced in foxf2a mutants and that the remaining pericytes display alterations in their morphologies. The authors further find that mutant animals can develop to adulthood but that in adult animals, both endothelial and pericyte morphologies are affected. They also show that mutant pericytes can partially repopulate the brain after genetic ablation.

Strengths:

The paper is well written and easy to follow. The authors now include pericyte marker gene analysis and solid quantifications of the observed phenotypes.

Weaknesses:

None left.

---

## [Referee Report · Reviewer #2 (Public review)]

Summary:

This study investigates the developmental and lifelong consequences of reduced foxf2 dosage in zebrafish, a gene associated with human stroke risk and cerebral small vessel disease (CSVD). The authors show that a ~50% reduction in foxf2 function through homozygous loss of foxf2a leads to a significant decrease in brain pericyte number, along with striking abnormalities in pericyte morphology-including enlarged soma and extended processes-during larval stages. These defects are not corrected over time but instead persist and worsen with age, ultimately affecting the surrounding endothelium. The study also makes an important contribution by characterizing pericyte behavior in wild-type zebrafish using a clever pericyte-specific Brainbow approach, revealing novel interactions such as pericyte process overlap not previously reported in mammals.

Strengths:

This work provides mechanistic insight into how subtle, developmental changes in mural cell biology and coverage of the vasculature can drive long-term vascular pathology. The authors make strong use of zebrafish imaging tools, including longitudinal analysis in transgenic lines to follow pericyte number and morphology over larval development and then applied tissue clearing and whole brain imaging at 3 and 11 months to further dissect the longitudinal effects of foxf2a loss. The ability to track individual pericytes in vivo reveals cell-intrinsic defects and process degeneration with high spatiotemporal resolution. Their use of a pericyte-specific Zebrabow line also allows, for the first time, detailed visualization of pericyte-pericyte interactions in the developing brain, highlighting structural features and behaviors that challenge existing models based on mouse studies. Together, these findings make the zebrafish a valuable model for studying the cellular dynamics of CSVD.

Weaknesses:

I originally suggested quantifying pericyte coverage across brain regions to address potential lineage-specific effects due to the distinct developmental origins of forebrain (neural crest-derived) and hindbrain (mesoderm-derived) pericytes. However, I appreciate the authors' response referencing recent work from their lab (Ahuja, 2024), which demonstrates that both neural crest and mesoderm contribute to pericyte lineages in the midbrain and hindbrain. The convergence of these lineages into a shared transcriptional state by 30 hpf, as shown by their single-cell RNA-seq data, makes it unlikely that regional quantification would provide meaningful lineage-specific insight. I agree with the authors that lineage tracing experiments often suffer from low sample sizes, and their updated findings challenge earlier compartmental models of pericyte origin. I therefore appreciate their rationale for not pursuing regional quantification and consider this concern addressed. Furthermore, my other two points regarding quantification of foxf2 levels and overall vascular changes have been thoroughly addressed in the revised manuscript. These additions significantly strengthen the paper's conclusions and improve the overall rigor of the study.

---

## [Referee Report · Reviewer #3 (Public review)]

Summary:

The goal of the work by Graff, et al. is to model CSVD in the zebrafish using foxf2a mutants. The mutants show loss of cerebral pericyte coverage that persists through adulthood, but it seems foxf2a does not regulate the regenerative capacity of these cells. The findings are interesting and build on previous work from the group. Limitations of the work include little mechanistic insight into how foxf2a alters pericyte recruitment/differentiation/survival/proliferation in this context, and the overlap of these studies with previous work in fox2a/b double mutants. However, the data analysis is clean and compelling and the findings will contribute to the field.

Comments on revisions:

The authors have addressed all of my original concerns.

---

## [Author Response]

The following is the authors’ response to the original reviews.

**eLife Assessment**
This study presents valuable findings that advance our understanding of mural cell dynamics and vascular pathology in a zebrafish model of cerebral small vessel disease. The authors provide compelling evidence that partial loss of foxf2 function leads to progressive, cell-intrinsic defects in pericytes and associated endothelial abnormalities across the lifespan, leveraging powerful in vivo imaging and genetic tools. The strength of evidence could be further improved by additional mechanistic insight and quantitative or lineage-tracing analyses to clarify how pericyte number and identity are affected in the mutant model.

Thank you to the reviewers for insightful comments and for the time spent reviewing the manuscript. We have strengthened the data through responding to the comments.

**Public Reviews:**

**Reviewer #1 (Public review):**
The paper by Graff et al. investigates the function of foxf2 in zebrafish to understand the progression of cerebral small vessel disease. The authors use a partial loss of foxf2 (zebrafish possess two foxf2 genes, foxf2a and foxf2b, and the authors mainly analyze homozygous mutants in foxf2a) to investigate the role of foxf2 signaling in regulating pericyte biology. They find that the number of pericytes is reduced in foxf2a mutants and that the remaining pericytes display alterations in their morphologies. The authors further find that mutant animals can develop to adulthood, but that in adult animals, both endothelial and pericyte morphologies are affected. They also show that mutant pericytes can partially repopulate the brain after genetic ablation.(1) Weaknesses: The results are mainly descriptive, and it is not clear how they will advance the field at their current state, given that a publication on mice has already examined the loss of foxf2 phenotype on pericyte biology (Reyahi, 2015, Dev. Cell).

The Reyahi paper was the earliest report of foxf2 mutant brain pericytes and remains illuminating. The work was very well technically executed. Our manuscript expands and at times, contradicts, their findings. We realized that we did not fully discuss this in our discussion, and this has now been updated. The biggest difference between the two studies is in the direction of change in pericytes after foxf2 knockout, a major finding in both papers. This is where it is important to understand the differences in methods. Reyahi et al., used a conditional knockout under Wnt1:Cre which will ablate pericytes derived from neural crest, but not those derived from mesoderm, nor will it affect foxf2 expression in endothelial cells. Our model is a full constitutive knockout of the gene in all brain pericytes and endothelial cells. For GOF, Reyahi used a transgenic model with a human FOXF2 BAC integrated into the mouse germline.

Both studies are important. We do not know enough about human phenotypes in patients with strokeassociated human FOXF2 SNVs to know the direction of change in pericyte numbers. We showed that the SNVs reduce FOXF2 gene expression in vitro (Ryu, 2022). Here we demonstrate dosage sensitivity in fish (showing phenotypes when 1 of 4 foxf2a + foxf2b alleles are lost, Figure 1F), supporting that slight reductions of FOXF2 in humans could lead to severe brain vessel phenotypes. For this reason, our work is complementary to the previously published work and suggests that future studies should focus on understanding the role of dosage, cell autonomy, and human pericyte phenotypes with respect to FOXF2. While some experiments are parallel in mouse and fish, we go further to look at cell death and regeneration, and to understand the consequences on the whole brain vasculature.

(2) Reyahi et al. showed that loss of foxf2 in mice leads to a marked downregulation of pdgfrb expression in perivascular cells. In contrast to expectation, perivascular cell numbers were higher in mutant animals, but these cells did not differentiate properly. The authors use a transgenic driver line expressing gal4 under the control of the pdgfrb promoter and observe a reduction in pericyte (pdgfrb-expressing) cells in foxf2a mutants. In light of the mouse data, this result might be due to a similar downregulation of pdgfrb expression in fish, which would lead to a downregulation of gal4 expression and hence reduced labelling of pericytes. The authors show a reduction of pdgfrb expression also in zebrafish in foxf2b mutants (Chauhan et al., The Lancet Neurology 2016).

Reyahi detected more pericytes in the Wnt1:Cre mouse, while we detected fewer in the foxf2a (and foxf2a;foxf2b) mutants. This may be because of different methods. For instance, because the mouse knockout is not a constitutive Foxf2 knockout, the observed increase in pericytes may be because mesodermal-derived pericytes proliferate more highly when the neural crest-derived pericytes are absent. Or does endothelial foxf2 activate pericyte proliferation when foxf2 is lost in some pericytes? It is also possible that mouse foxf2 has a different role from its fish ortholog. Despite these differences, there are common conclusions from both models. For instance, both mouse and fish show foxf2 controls capillary pericyte numbers, albeit in different directions. Both show hemorrhage and loss of vascular stability as a result. Both papers identify the developmental window as critical for setting up the correct numbers of pericytes.

As the reviewer suggested, it was important to test whether pdgfrb is downregulated in fish as it is in mice. To do this, we measured expression of *pdgfrb* in *foxf2* mutants using hybridization chain reaction (HCR) of *pdgfrb* in *foxf2* mutants. The results show no change in *pdgfrb* mRNA in *foxf2a* mutants at two independent experiments (Fig S3). Independently, we integrated *pdgfrb* transgene intensity (using a single allele of the transgene so there are no dose effects) in *foxf2a* mutants vs. wildtype. We found no difference (Fig S3) suggesting that *pdgfrb* is a reliable reporter for counting pericytes in the foxf2a knockout. The reviewer is correct that we previously showed downregulation of *pdgfrb* in *foxf2b* mutants at 4 dpf using colorimetric ISH. *foxf2a* and *foxf2b* are unlinked, independent genes (~400 M years apart in evolution) and may have different regulation.

(3) It would be important to clarify whether, also in zebrafish, foxf2a/foxf2b mutants have reduced or augmented numbers of perivascular cells and how this compares to the data in the mouse.

We discuss methodological differences between Reyahi and our work in point (1) above. The reduction in pericytes in foxf2a;foxf2b mutants has been previously published (Ryu, 2022, Supplemental Figure 1) and shown again here in Supplemental Figure 2. Numbers are reduced in double mutants up to 10 dpf, suggesting no recovery. Further, in response to reviewer comments, we have quantified pericytes in the whole fish brain (Figure 3E-G) and show reduced pericytes in the adult, reduced vessel network length, and importantly that the pericyte density is reduced. In aggregate, our data shows pericyte reduction at 5 developmental stages from embryo through adult. The reason for different results from the mouse is unknown and may reflect a technical difference (constitutive vs Wnt1:Cre) or a species difference.

(4) The authors should perform additional characterization of perivascular cells using marker gene expression (for a list of markers, see e.g., Shih et al. Development 2021) and/or genetic lineage tracing.

This is a good point. We have added HCR analysis of additional markers. Results show co-expression of foxf2a, foxf2b, nduf4la2 and pdgfrb in brain pericytes (Fig 2, Fig S3).

(5) The authors motivate using foxf2a mutants as a model of reduced foxf2 dosage, "similar to human heterozygous loss of FOXF2". However, it is not clear how the different foxf2 genes in zebrafish interact with each other transcriptionally. Is there upregulation of foxf2b in foxf2a mutants and vice versa? This is important to consider, as Reyahi et al. showed that foxf2 gene dosage in mice appears to be important, with an increase in foxf2 gene dosage (through transgene expression) leading to a reduction in perivascular cell numbers.

We agree that dosage is a very important concept and show phenotypes in foxf2a heterozygotes (Fig 1F). To test the potential compensation from foxf2b, we have added qPCR for foxf2b in foxf2a mutants as well as HCR of foxf2b in foxf2a mutants (Fig S3C,D). There is no change in *foxf2b* expression in foxf2a mutants. We discuss dosage in our discussion.

(6) Figures 3 and 4 lack data quantification. The authors describe the existence of vascular defects in adult fish, but no quantifiable parameters or quantifications are provided. This needs to be added.

This query was technically challenging to address, but very worthwhile. We have not seen published methods for quantifying brain pericytes along with the vascular network (certainly not in zebrafish adults), so we developed new methods of analyzing whole brain vascular parameters of cleared adult brains (Figure S6) using a combination of segmentation methods for pericytes, endothelium and smooth muscle. We have added another author (David Elliott) as he was instrumental in designing methods. We find a significant decrease in vessel network length in foxf2a mutants at 3 month and 6 months (Figures 3F and 4G). Similarly, we show a lower number of brain pericytes in foxf2a mutants (Figure 3E). Finally, we added whole brain analysis of smooth muscle coverage (Figure 4) and show no change in vSMC number or coverage of vessels at 5 and 10 dpf or adult, respectively, pointing to pericytes being the cells most affected. Thank you, this query pushed us in a very productive direction. These methods will be extremely useful in the future!

(7) The analysis of pericyte phenotypes and morphologies is not clear. On page 6, the authors state: "In the wildtype brain, adult pericytes have a clear oblong cell body with long, slender primary processes that extend from the cytoplasm with secondary processes that wrap around the circumference of the blood vessel." Further down on the same page, the authors note: "In wildtype adult brains, we identified three subtypes of pericytes, ensheathing, mesh and thin-strand, previously characterized in murine models." In conclusion, not all pericytes have long, slender primary processes, but there are at least three different sub-types? Did the authors analyze how they might be distributed along different branch orders of the vasculature, as they are in the mouse?

We have reworded the text on page 5/6 to be clearer that embryonic pericytes are thin strand only. Additional pericyte subtypes develop later are seen in the mature vasculature of the adult. We could not find a way to accurately analyze pericyte subtypes in the adult brain. The imaging analysis to count pericytes used soma as machine learning algorithms have been developed to count nuclei but not analyze processes.

(8) Which type of pericyte is affected in foxf2a mutant animals? Can the authors identify the branch order of the vasculature for both wildtype and mutant animals and compare which subtype of pericyte might be most affected? Are all subtypes of pericytes similarly affected in mutant animals? There also seems to be a reduction in smooth muscle cell coverage.

Please see the response to (7) about pericyte subtypes. In response to the reviewer’s query, we have now analyzed vSMCs in the embryonic and adult brain. In the embryonic brain we see no statistical differences in vSMC number at 5 and 10 dpf (Figure 4). In the adult, vSMC length (total length of vSMCs in a brain) and vSMC coverage (proportion of brain vessels with vSMCs) are not significantly different. This data is important because it suggests that foxf2a has a more important role in pericytes than in vSMCs.

(9) Regarding pericyte regeneration data (Figure 7): Are the values in Figure 7D not significantly different from each other (no significance given)?

Any graphs missing bars have no significance and were left off for clarity. We have stated this in the statistical methods.

(10) In the discussion, the authors state that "pericyte processes have not been studied in zebrafish".

Ando et al. (Development 2016) studied pericyte processes in early zebrafish embryos, and Leonard et al. (Development 2022) studied zebrafish pericytes and their processes in the developing fin. We apologize, this was not meant to say that pericyte processes had not been studied before, we have reworded this to make clear the intent of the sentence. We were trying to emphasize that we are the first to quantify processes at different stages, especially in foxf2 mutants. Processes change morphology over development, especially after 5 dpf, something that our data captures. Our images are of stages that have not been previously characterized. We added a reference to Mae et al., who found similar process length changes in a mouse knockout of a different gene, and to Leonard who previously showed overlap of processes in a different context in fish.

**Reviewer #2 (Public review):**
Summary:This study investigates the developmental and lifelong consequences of reduced foxf2 dosage in zebrafish, a gene associated with human stroke risk and cerebral small vessel disease (CSVD). The authors show that a ~50% reduction in foxf2 function through homozygous loss of foxf2a leads to a significant decrease in brain pericyte number, along with striking abnormalities in pericyte morphologyincluding enlarged soma and extended processes-during larval stages. These defects are not corrected over time but instead persist and worsen with age, ultimately affecting the surrounding endothelium. The study also makes an important contribution by characterizing pericyte behavior in wild-type zebrafish using a clever pericyte-specific Brainbow approach, revealing novel interactions such as pericyte process overlap not previously reported in mammals.Strengths:This work provides mechanistic insight into how subtle, developmental changes in mural cell biology and coverage of the vasculature can drive long-term vascular pathology. The authors make strong use of zebrafish imaging tools, including longitudinal analysis in transgenic lines to follow pericyte number and morphology over larval development, and then applied tissue clearing and whole brain imaging at 3 and 11 months to further dissect the longitudinal effects of foxf2a loss. The ability to track individual pericytes in vivo reveals cell-intrinsic defects and process degeneration with high spatiotemporal resolution. Their use of a pericyte-specific Zebrabow line also allows, for the first time, detailed visualization of pericytepericyte interactions in the developing brain, highlighting structural features and behaviors that challenge existing models based on mouse studies. Together, these findings make the zebrafish a valuable model for studying the cellular dynamics of CSVD.Weaknesses:(11) While the findings are compelling, several aspects could be strengthened. First, quantifying pericyte coverage across distinct brain regions (forebrain, midbrain, hindbrain) would clarify whether foxf2a loss differentially impacts specific pericyte lineages, given known regional differences in developmental origin, with forebrain pericytes being neural crest-derived and hindbrain pericytes being mesoderm-derived.

In recently published work from our lab, we published that both neural crest and mesodermal cells contribute to pericytes in both the mid and hindbrain, and could not confirm earlier work suggesting more rigid compartmental origins (Ahuja, 2024). In the Ahuja, 2024 paper we noted that lineage experiments are often limited by n’s which is why this may not have been discovered before. This makes us skeptical that counting different regions will allow us to interpret data about neural crest and mesoderm. Further, Ahuja 2024 shows that pericyte intermediate progenitors from both mesoderm and neural crest are indistinguishable at 30 hpf through single cell sequencing and have converged on a common phenotype.

(12) Second, measuring foxf2b expression in foxf2a mutants would better support the interpretation that total FOXF2 dosage is reduced in a graded fashion in heterozygote and homozygote foxf2a mutants.

We have done both qPCR for foxf2b in foxf2a mutants and HCR (quantitative ISH). This is now reported in Fig S3.

(13) Finally, quantifying vascular density in adult mutants would help determine whether observed endothelial changes are a downstream consequence of prolonged pericyte loss. Correlating these vascular changes with local pericyte depletion would also help clarify causality.

We have added this data to Figure 3 and 4. Please also see response (6).

**Reviewer #3 (Public review):**
Summary:The goal of the work by Graff et al. is to model CSVD in the zebrafish using foxf2a mutants. The mutants show loss of cerebral pericyte coverage that persists through adulthood, but it seems foxf2a does not regulate the regenerative capacity of these cells. The findings are interesting and build on previous work from the group. Limitations of the work include little mechanistic insight into how foxf2a alters pericyte recruitment/differentiation/survival/proliferation in this context, and the overlap of these studies with previous work in fox2a/b double mutants. However, the data analysis is clean and compelling, and the findings will contribute to the field.(14) Please make Figures 5C and 5E red-green colorblind friendly.

Thank you. We have changed the colors to light blue and yellow to be colorblind friendly.

**Reviewer #3 (Recommendations for the authors):**
(15) I'm not sure this reviewer totally agrees with the assessment that foxf2a loss of function, while foxf2b remains normal, is the same as FOXF2 heterozygous loss of function in humans. The discussion of the gene dosage needs to be better framed, and the authors should carry out qPCR to show that foxf2b levels are not altered in the foxf2a mutant background.

We have added data on foxf2b expression in foxf2a mutants to Fig S3. We have updated the results.

(16) Figure 4/SF7- is the aneurysm phenotype derived from the ECs or pericytes? Cell-type-specific rescues would be interesting to determine if phenotypes are rescued, especially the developmental phenotypes (it is appreciated that carrying out rescue experiments until adulthood is complex). When is the earliest time point that aneurysm-like structures are seen?

This is a fascinating question, especially as we show that endothelial cells (vessel network length) are affected in the adult mutants. The foxf2a mutants that we work with here are constitutive knockouts. While a strategy to rescue foxf2a in specific lineages is being developed in the laboratory this will require a multi-generation breeding effort to get drivers, transgenes and mutants on the same background, and these fish are not currently available. Thank you for this comment- it is something we want to follow up on.

(17) Figure 5 - This is very nice analysis.

Thank you! We think it is informative too.

(18) Figure 6 - needs to contain control images

We have added wildtype images to figure 6A.

(19) Figure 7- vessel images should be shown to demonstrate the specificity of NTR treatment to the pericytes.

We have added the vessel images to Figure 7. We apologize for the omission.